# Self-supervised Representation Learning from Random Data Projectors

**Yi Sui**
Layer 6 AI
amy@layer6.ai

**Tongzi Wu**
Layer 6 AI
tongzi@layer6.ai

**Jesse C. Cresswell**
Layer 6 AI
jesse@layer6.ai

**Ga Wu**
Dalhousie University
ga.wu@dal.ca

**George Stein**
Layer 6 AI
george@layer6.ai

**Xiao Shi Huang**
Layer 6 AI
gary@layer6.ai

**Xiaochen Zhang**
Layer 6 AI
lisa@layer6.ai

**Maksims Volkovs**
Layer 6 AI
maks@layer6.ai

## Abstract

Self-supervised representation learning (SSRL) has advanced considerably by exploiting the transformation invariance assumption under artificially designed data augmentations. While augmentation-based SSRL algorithms push the boundaries of performance in computer vision and natural language processing, they are often not directly applicable to other data modalities, and can conflict with application-specific data augmentation constraints. This paper presents an SSRL approach that can be applied to *any* data modality and network architecture because it does not rely on augmentations or masking. Specifically, we show that high-quality data representations can be learned by reconstructing random data projections. We evaluate the proposed approach on a wide range of representation learning tasks that span diverse modalities and real-world applications. We show that it outperforms multiple state-of-the-art SSRL baselines. Due to its wide applicability and strong empirical results, we argue that learning from randomness is a fruitful research direction worthy of attention and further study.

## 1 Introduction

Learning data representations in a self-supervised manner is commonly associated with well-designed pretext tasks that can create heuristics for machine learning models to identify and encode useful information from unlabelled data. While pretext tasks are often highly customized to specific applications, they are typically based on a handful of underlying assumptions. Pretext tasks utilizing the transformation invariance assumption across data augmentation views show leading performance in multiple research domains. For computer vision, image representations are commonly guided to remain identical after cropping, rotating, flipping, or corrupting, among others (Chen et al., 2020; Grill et al., 2020). Similarly, in natural language processing, sentences with similar words and semantic meaning are expected to have the same representation (Wei & Zou, 2019; Wu et al., 2019). In these and other domains transformation invariance is encouraged explicitly through contrastive or momentum objectives that aim to bring together representations before and after transformation.

Despite their strong performance in many domains, self-supervised representation learning (SSRL) algorithms that enforce transformation invariance are limited in that they do not support generic data types. Many well-known data augmentation methods are tailored to specific modalities, and are restricted in their generality across different domains. For instance, Gaussian noise corruption cannot be applied directly to textual data, while image rotation is not applicable in the tabular domain. Thus, augmentation based SSRL techniques are inherently constrained in their cross-domain applicability.

While these examples show modality limitations, a more subtle problem is that even standard augmentations can conflict with application-specific constraints. For example, pathology images of stained tissue samples have low color variation (Shen et al., 2022; Kang et al., 2023), so naïve color-jittering produces unnatural augmentations (see Fig. 1). Such incorrect images may be inappropriate and unsafe for use in critical medical imaging applications (Elgendi et al., 2021). For time series with high periodicity, such as online monitoring data, random shifting augmentations can create

identical augmented views that are not useful for pretext tasks (Eldele et al., 2021; Zhang & Ma, 2022). In tabular settings where relatively few options for augmentations are available, random noise addition or random swapping of features between training examples can easily produce unrealistic examples. As a concrete example from particle physics, consider energies and momenta of interacting particles collected experimentally (Brehmer & Cranmer, 2020). Since the incoming and outgoing energies are constrained by the laws of physics, only certain combinations are possible to observe. Tabular augmentations would produce unphysical combinations, harming the consistency of learned representations and leading to unpredictable failures on downstream tasks (Shorten & Khoshgoftaar, 2019).

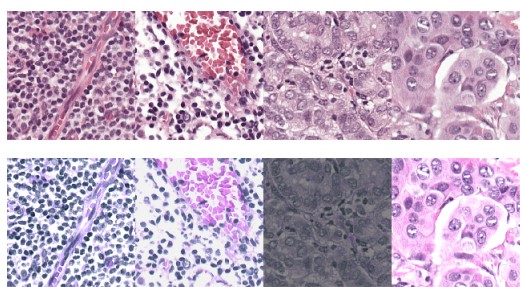

Figure 1: **Top:** H&E stained histopathology images have a characteristic appearance with blue tones indicating cell nuclei, while cytoplasm is stained pink (Chan, 2014). **Bottom:** Color jitter with the standard settings of (Chen & He, 2021) produces unrealistic augmentations with altered meanings. Choosing good augmentations requires domain knowledge (Shen et al., 2022).

Instead of relying on transformation invariance, many self-supervised learning techniques involve masking and reconstructing input data including masked image modelling (He et al., 2022; Xie et al., 2022) for computer vision and masked language modelling (Devlin et al., 2019; Raffel et al., 2020) for natural language processing. However, these methods often require specific backbone architectures like transformers (Vaswani et al., 2017) to achieve good performance (Balestriero et al., 2023).

In this paper, we introduce a SSRL training scheme that requires neither domain-specific data augmentations nor particular architectures as in masking approaches. Instead, the proposed learning method is based on a surprising hypothesis that good data representations can be obtained by learning to simultaneously reconstruct multiple randomly generated data projection functions. The hypothesis comes from the conventional motivation of representation learning – capturing and extracting abstract and valuable concepts that can support a range of downstream predictive tasks (Bengio et al., 2013; Le-Khac et al., 2020). In particular, the downstream tasks could include arbitrary data projections. Formally, given random projection functions $G = \{\cdots g^{(k)}(\mathbf{x}) \cdots\}$ whose input domains are raw data features $\mathbf{x} \in \mathcal{X}$, the motivation above suggests that, for a good representation $\mathbf{z}$ of data $\mathbf{x}$, there is another group of simple prediction functions $H = \{\cdots h^{(k)}(\mathbf{z}) \cdots\}$ that can correctly predict the random functions' outputs. With this insight, the representation learning task can be construed as a search for a combination of representation $\mathbf{z}$ and prediction functions $H$ that can reproduce random data projections. In short, we conduct SSRL by *learning from randomness* (LFR).

The primary advantage of LFR is that random projection functions $G$ can easily be created for *any* data modality. One straightforward instantiation is by taking a neural network with suitable architecture for consuming the data, and randomly initializing its parameters. Hence, LFR applies to all subfields of SSRL. Also, data augmentations are not used, so LFR avoids any concerns of unsafe, identical, or unrealistic augmentations as discussed above.

We empirically evaluate the effectiveness of LFR on a wide range of representation learning tasks that span diverse data types (including image, sequential, and tabular) and multiple real-world applications (banking, healthcare and natural sciences). The results show that LFR outperforms commonly used domain-agnostic SSRL algorithms. It is even competitive with many domain-specific approaches that rely heavily on expert knowledge for their data augmentation designs. The remarkable performance demonstrates that learning high-quality data representations from randomness is a feasible and plausible alternative when the transformation invariance assumption is hard to establish or enforce in a given application domain.

## 2 BACKGROUND AND RELATED WORK

Self-supervised representation learning (SSRL) methods enable the extraction of informative and compact representations from raw data without manual annotation or labelling. These methods rely on large amounts of unlabeled data and pretext tasks to implicitly model the observed distribution

and optimize deep neural networks. In computer vision (CV) and natural language processing (NLP), SSRL has gained considerable attention due to an abundance of widely available unlabeled data.

Adopting the definitions as outlined in (Balestriero et al., 2023), SSRL methods in the field of CV can be grouped into four main categories: deep metric learning (Chen et al., 2020), self-distillation (Grill et al., 2020; Chen et al., 2020), canonical correlation analysis (Zbontar et al., 2021), and masked image modeling (He et al., 2022). Among these, the first three rely on creating positive views of a given image to learn invariances. This is achieved by creating augmented versions of the same image and enforcing that the latent embeddings of these versions should be identical, with the underlying assumption that the semantic meaning of the original image is invariant across different views. Recently, masked image modeling has emerged as a popular SSRL approach due to the success of Vision Transformers (Dosovitskiy et al., 2021). These methods predict masked vision tokens (He et al., 2022) or pixel patches (Doersch et al., 2015), which have proven useful in learning representations. In NLP, masked language modelling is very prominent. The dominant models, including BERTs and GPTs, are trained to predict masked language tokens, which encourages the model to encode contextual information and reconstruct its inputs. Occasionally, other efficient language-focused pretext tasks emerge in the literature, such as maximizing the mutual information between a global sentence representation and $n$-grams in the sentence (Kong et al., 2020).

Despite the remarkable success in CV and NLP, the effectiveness of SSRL for other data modalities, such as tabular data, has been limited (Bahri et al., 2022; Balestriero et al., 2023). One challenge for SSRL methods relying on transformation invariance lies in identifying and designing appropriate augmentations. Augmentation strategies that work well for one modality may not directly translate to others due to inherent differences, and the choice of suitable augmentations can also be influenced by the specific application domain. For example, augmentations designed for natural images may not be suitable for medical images with low color variation, leading to unrealistic results and unsatisfactory performance (Kang et al., 2023). Appendix A provides further information on research efforts dedicated to designing effective modality- and application-specific augmentation techniques. While masking approaches offer general applicability to all data modalities, the most effective frameworks often rely on transformer-based backbones for optimal performance (Majmundar et al., 2022; He et al., 2022; Cheng et al., 2023). In this work, we focus on a model-agnostic SSRL approaches. Classic autoencoder-based methods provide an alternative to SSRL without relying explicitly on transformation invariance (Hinton & Salakhutdinov, 2006; Vincent et al., 2008; Zhang et al., 2017). However, these methods tend to prioritize low-level reconstruction over capturing high-level abstractions required for downstream tasks, resulting in suboptimal performance in practical applications (Liu et al., 2021).

The current landscape of SSRL research highlights the need for a more versatile and effective approach capable of addressing a wider range of modalities, applications, and architectures.

## 3 REPRESENTATION LEARNING FROM RANDOM DATA PROJECTORS

In this section, we present *learning from randomness* (LFR), an efficient and general SSRL algorithm. We recap the representation learning problem setting as the following: given observed raw data $X = \{\cdots \mathbf{x}_i \cdots\}$, where all data points share the same feature domain $\mathcal{X}$, the representation learning task is to learn a function $f_\theta(\mathcal{X})$ that produces a low-dimensional representation $\mathbf{z}_i \in \mathcal{Z}$ for each raw data input $\mathbf{x}_i$. The representation $\mathbf{z}_i$ should carry useful information about $\mathbf{x}_i$ such that for an arbitrary downstream task $g(\mathcal{X})$ it is possible to learn a simple prediction function $h_\phi(\mathcal{Z})$ that replicates $g(\mathbf{x}_i)$ as $h_\phi(f_\theta(\mathbf{x}_i))$ for all $\mathbf{x}_i \in \mathcal{X}$.

### 3.1 PRETEXT TASK: MULTI-OBJECTIVE LEARNING FROM RANDOMNESS

As mentioned in the problem statement above, the ultimate purpose of representation learning is to support arbitrary downstream predictive tasks. In reality, there is usually a small subset of downstream tasks which are considered important. It is not *a priori* clear that directly learning to predict purely random tasks would lead to good representations for important tasks.

To demonstrate the possibility of learning from randomness, we propose the surprising pretext task shown in Figure 2. The pretext task contains three components, namely a representation model $f_\theta(\mathcal{X})$, a set of randomly generated data projection functions $G = \{\cdots g^{(k)}(\mathcal{X}) \cdots\}$, and a set of

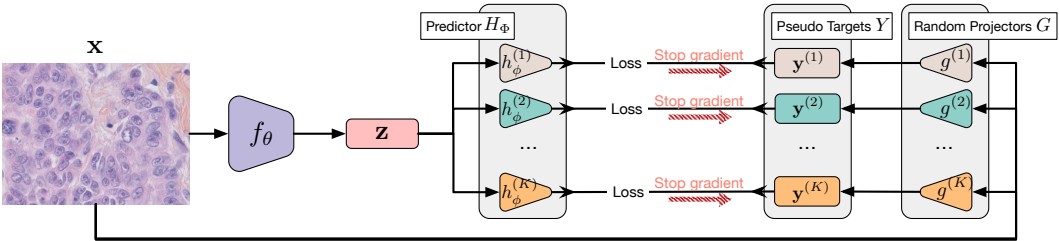

Figure 2: Our proposed architecture for learning from randomness. An input $\mathbf{x}$ is encoded by $f_\theta$ into a useful representation $\mathbf{z}$, while also being fed to random projection functions $g^{(k)}$. Simple, learnable predictor functions $h_\phi^{(k)}$ try to match the outputs $\mathbf{y}^{(k)}$ from the projectors $g^{(k)}$, which is only possible when $\mathbf{z}$ contains rich information about the input.

simple predictors $H_\Phi = \{\cdots h_\phi^{(k)}(\mathcal{Z})\cdots\}$ that aim to predict the outcome of each random projection function respectively. Formally, we propose optimizing the high-level objective:

$$\underset{\theta,\Phi}{\arg\min} \sum_{\mathbf{x}_i \in \mathcal{X}} \sum_k \mathcal{D}\left[g^{(k)}(\mathbf{x}_i), h_\phi^{(k)}\left(f_\theta(\mathbf{x}_i)\right)\right] + \lambda_1 \Omega_1(\theta) + \lambda_2 \Omega_2(\Phi), \tag{1}$$

where $\mathcal{D}[\cdot,\cdot]$ is a divergence metric measuring the similarity between its inputs, the $\Omega(\cdot)$'s denote regularization terms, and the $\lambda$'s are the corresponding weights. To make this a non-trivial task, the predictors $h_\phi^{(k)}$ should have limited capacity, such as being linear functions, or simple neural networks with a few layers. This objective aligns with existing SSRL methods that use predictors (Grill et al., 2020; Chen & He, 2021).

Objective (1) is essentially a lower-bound of the maximum likelihood estimation (MLE) objective; we aim to maximize the probability of observing data projections $\mathbf{y}_i^{(k)} = g^{(k)}(\mathbf{x}_i)$ given all datapoints,

$$\begin{aligned}
\sum_i \log p\left(Y_i \mid \mathbf{x}_i\right) &= \sum_i \sum_k \log \int_{\mathbf{z}_i} p\left(\mathbf{y}_i^{(k)} \mid \mathbf{z}_i\right) p\left(\mathbf{z}_i \mid \mathbf{x}_i\right) d\mathbf{z}_i \\
&\geq \sum_i \sum_k \int_{\mathbf{z}_i} p\left(\mathbf{z}_i \mid \mathbf{x}_i\right) \log p\left(\mathbf{y}_i^{(k)} \mid \mathbf{z}_i\right) d\mathbf{z}_i = \sum_i \sum_k \log p\left(\mathbf{y}_i^{(k)} \mid \mathbf{z}_i = f_\theta(\mathbf{x}_i)\right),
\end{aligned} \tag{2}$$

where $p\left(\mathbf{z}_i \mid \mathbf{x}_i\right)$ is a Dirac delta distribution since the representation model is a deterministic function. As an example of the connection between Objectives (1) and (2), when the $p\left(\mathbf{y}_i^{(k)} \mid \mathbf{z}_i\right)$ are assumed to be Gaussian distributions, the corresponding $\mathcal{D}$ in Objective (1) is the Euclidean distance. We discuss other options for $\mathcal{D}$ below.

A straightforward training strategy for Objective (1) is joint training where we treat the representation model $f_\theta$ and predictors $H_\Phi$ as a multi-objective autoencoder, updating all parameters in the same backpropagation pass. However, in preliminary experiments this naïve training strategy showed fluctuating progress which prevented the model from converging to satisfactory solutions.

We tackle the training instability issue by adopting the classic Expectation-Maximization (EM) method. Considering the MLE lower-bound (Objective (2)), optimizing the representation model $f_\theta$ is an E-step that repositions the posterior distribution of data $p(\mathcal{Z}|\mathcal{X})$ given fixed log-likelihood estimation modules $\log p\left(\mathbf{y}_i^{(k)} \mid \mathbf{z}_i\right)$. For the M-step, the representation distribution is fixed and we optimize the predictor heads $H_\Phi$. Details of the derivation can be found in Appendix G. Hence, in this work we train the proposed SSRL model by alternating steps:

**E-step:** Optimize the representation model parameters $\theta$ for one iteration

$$\underset{\theta}{\arg\min} \sum_i \sum_k \mathcal{D}\left[g^{(k)}(\mathbf{x}_i), h_\phi^{(k)}(f_\theta(\mathbf{x}_i))\right] + \lambda_1 \Omega_1(\theta), \tag{3}$$

**M-step:** Optimize the predictor model parameters $\Phi$ for $M$ iterations

$$\underset{\Phi}{\arg\min} \sum_i \sum_k \mathcal{D}\left[g^{(k)}(\mathbf{x}_i), h_\phi^{(k)}(f_\theta(\mathbf{x}_i))\right] + \lambda_2 \Omega_2(\Phi). \tag{4}$$

Optimizing the predictor model for more iterations than the representation model brings the predictor closer to its optimal performance with the latest representation model $f_\theta$. Previous studies have shown that optimizing the predictor to achieve optimality leads to improved performance (Chen & He, 2021; Grill et al., 2020; Tian et al., 2021).

## 3.2 DIVERGENCE MEASURE: BATCH-WISE BARLOW TWINS

Now we return to options for the divergence $\mathcal{D}$ in Objective (1). While there are several common choices of divergence used in machine learning such as Mean Squared Error (MSE), Cross Entropy (CE), or the Contrastive (Chopra et al., 2005) and Triplet (Schroff et al., 2015) losses, they are often inadequate for enforcing identifications between subtly different points as is crucial for representation learning tasks; MSE downweights the importance of small errors, CE is ill suited for regression tasks, while the Contrastive and Triplet losses introduce significant stochasticity.

The Barlow Twins loss (Zbontar et al., 2021) has garnered much interest in the SSRL literature as it disentangles learned representations through redundancy reduction. We also note its ability to scale to very high-dimensional vectors (Zbontar et al., 2021). Thus, we introduce Batch-wise Barlow Twins (BBT), a variant that measures representation differences between data instances from two sources, the random projector $g^{(k)}$ and the predictor $h_\phi^{(k)}$, rather than disentangling the representation encoding. We define the BBT loss as

$$\mathcal{L}_{\text{BBT}} = \sum_k \sum_i \left[ \left(1 - c_{ii}^{(k)}\right)^2 + \lambda \sum_{j \neq i} c_{ij}^{(k)^2} \right], \tag{5}$$

where the $c_{ij}$ are the entries of a cosine similarity matrix,

$$c_{ij}^{(k)} = \frac{\mathbf{y}_i^{(k)^\top} \hat{\mathbf{y}}_j^{(k)}}{\left\| \mathbf{y}_i^{(k)} \right\|_2 \left\| \hat{\mathbf{y}}_j^{(k)} \right\|_2}, \quad \mathbf{y}_i^{(k)} = g^{(k)}(\mathbf{x}_i), \quad \hat{\mathbf{y}}_i^{(k)} = h_\phi^{(k)} \left( f_\theta(\mathbf{x}_i) \right), \tag{6}$$

and $\mathbf{y}_i^{(k)}, \hat{\mathbf{y}}_i^{(k)} \in \mathbb{R}^{d^{(k)}}$. Compared to the loss in (Zbontar et al., 2021), Equation (5) has an extra summation over the ensemble $k$. The main difference comes from the definition of the cosine similarity matrix; our cosine similarity is an $m \times m$ matrix with $m$ the batch size, whereas in Barlow Twins it is a $d^{(k)} \times d^{(k)}$ matrix.

## 3.3 DIVERSITY ENCOURAGEMENT ON RANDOM DATA PROJECTORS

*Learning from randomness* aims to extract useful representations from random projection functions $g^{(k)}(\mathcal{X}) \in G$ which mimic arbitrary downstream tasks. In practice we create multiple data projections by randomly initializing neural networks that reuse the architecture design of $f_\theta$, but scaled down, which avoids the need to make domain-specific choices about the projectors. Functions generated this way can often capture similar information to each other when diversity is not specifically encouraged, which limits the generalization capabilities of the representations learned by $f_\theta$. While increasing the number of random projection functions could mitigate the diversity problem by brute force, such an approach is computationally wasteful because it would maintain many similar projectors.

We propose a solution that picks $K$ diverse projectors from $N \gg K$ randomly generated candidates. The underlying hypothesis is that one sufficiently large batch of data can reveal the behavioral differences between candidate random projectors. Presuming there is a batch of data $X \in \mathbb{R}^{m \times d}$, for each of the $N$ randomly generated projectors $g^{(k)}(\mathcal{X}) \in G$ we produce the normalized outputs

$$Y^{(k)} = g^{(k)}(X)/\|g^{(k)}(X)\|_2, \quad Y^{(k)} \in \mathbb{R}^{m \times d^{(k)}}. \tag{7}$$

We then compute the cosine similarity over the batch of outputs for each projector as

$$A^{(k)} = Y^{(k)}(Y^{(k)})^\top, \quad A^{(k)} \in \mathbb{R}^{m \times m}. \tag{8}$$

By flattening the matrix $A^{(k)}$ and again normalizing, we obtain a vector $\mathbf{a}^{(k)} \in \mathbb{R}^{m^2 \times 1}$, which acts as the signature of the $k$'th projector with respect to the batch. Finally, to select $K$ target models from

the $N$ candidates, we search for a subset that maximizes the following binary constraint optimization problem involving matrices $\tilde{A}$ made from $K$ stacked vectors $\mathbf{a}^{(k)}$,

$$\operatorname*{argmax}_{\mathbf{s}} |\det(B)| \quad \text{s.t.} \quad B = \tilde{A}\tilde{A}^{\top}, \quad \tilde{A} = \left[ \mathbf{a}^{(k)} \,\middle|\, k \in [0, N], s_k = 1, \sum_{k'} s_{k'} = K \right], \quad (9)$$

where the 1's in the binary vector $\mathbf{s} \in \{0, 1\}^N$ indicate the chosen projectors. While this problem is known to be NP-hard, approximate methods such as the Fast Determinantal Point Process (Chen et al., 2018) can find good solutions in reasonable time. It is worth noting that our diversity encouragement solution does not involve gradient computations, and can be run once as a pre-processing step without occupying computation resources during the SSRL training phase. In addition, we also explored other diversity encouraging techniques through initialization in our experiments and provide analysis of the impact on performance in Section 4.5. We summarize the full LFR algorithm in Appendix B.

## 4 EXPERIMENTS AND EVALUATION

### 4.1 DATASETS

We consider diverse data types to show the wide-ranging applicability of *learning from randomness*.

**Time series:** We utilized two standard time-series datasets, Human Activity Recognition (HAR) (Anguita et al., 2013) and Epileptic Seizure Recognition (Epilepsy) (Andrzejak et al., 2001). Both datasets were pre-processed using the same methods as in TS-TCC (Eldele et al., 2021). As a larger scale test we also include the MIMIC-III dataset, a standard in the medical domain for tasks involving electronic health record data. We utilized the pre-processed version of the MIMIC-III Benchmark dataset (Harutyunyan et al., 2019), and focused on the length-of-stay task (Yèche et al., 2021) which is framed as a 10-class classification problem, where each class represents a different duration of stay.

**Tabular:** We used three tabular UCI datasets in our experiments: Adult Income (Income) (Kohavi, 1996), First Order Theorem Proving (Theorem) (Bridge et al., 2014), and HEPMASS (Baldi et al., 2016). For Income, a binary classification problem, we followed the data preprocessing steps in (Ucar et al., 2021). The Theorem dataset is framed a a 6-class classification problem. The much larger HEPMASS dataset is another binary classification task which includes 7 million training and 3.5 million testing events, each with 27 features.

**Computer vision:** We tested on Kvasir (Pogorelov et al., 2017), a medical image dataset consisting of 8,000 images of the gastrointestinal tract. There are eight balanced classes including seven disease types as well as healthy images. We followed (Kalra et al., 2023) to resize all images to $100 \times 80$ pixels and split the data into 6,000 images for training and 2,000 for testing. We provide further results on CIFAR10 in Appendix D.6.

### 4.2 IMPLEMENTATIONS

**Evaluation:** All the downstream tasks in our study are treated as classification problems. To evaluate the quality of the pre-trained representations, we employed supervised classifiers that are specific to each dataset. For the MIMIC-III dataset we utilized a MLP classifier (Yèche et al., 2021). For tabular datasets, we used logistic regression, similar to the approach in STab (Hajiramezanali et al., 2022). For the remaining datasets, a linear classifier was employed. The classifiers were trained on the frozen representations of the training set and evaluated on the test set, which is the most commonly used protocol. We also include finetuning results in Appendix D.4 and transfer learning results on Time Series in Appendix D.1. Accuracy is our primary evaluation metric, except for MIMIC-III where we adopted linearly weighted Cohen's Kappa as in (Yèche et al., 2021), with higher values indicating better agreement. To ensure the robustness of our results, we conducted multiple random runs and report the mean and standard deviation, using 5 runs for tabular datasets and 3 runs for others.

**Model architectures:** Regarding the model architectures, we adopted similar backbone encoders as previous works. For the HAR and Epilepsy datasets, we utilized the same 3-block convolutional layers as TS-TCC (Wang et al., 2017). For the MIMIC-III dataset, we employed the Temporal Convolutional Network used by NCL (Yèche et al., 2021). For the Tabular datasets, we used 4-layer MLPs, following the approach in SCARF (Bahri et al., 2022). For Kvasir, we employed the ResNet18

Table 1: Baseline methods

| Category | Method | Description |
|---|---|---|
| Domain-agnostic | Autoencoder (Rumelhart et al., 1986) | Encoder/decoder with low dimensional latents trained via the reconstruction loss. |
| | DACL (Verma et al., 2021) | Self-supervised learning method that uses mix-up as data augmentation across modalities. |
| | DIET (Balestriero, 2023) | Self-supervised learning method that predicts the datum index as a pretext task. |
| Domain-specific augmentations | SimCLR (Chen et al., 2020) | Contrastive learning method with both positive and negative pairs. |
| | SimSiam (Chen & He, 2021) | Self-supervised learning with Siamese networks and only positive pairs. |
| Time series | TS-TCC (Eldele et al., 2021) | Contrastive learning method that uses a correlation-based similarity to capture temporal relationships, and time-series augmentations to generate positive and negative views. |
| Tabular | SCARF (Bahri et al., 2022) | Adaptation of SimCLR to tabular domains, using random corruption for dual views. |
| | STab (Hajiramezanali et al., 2022) | An augmentation-free framework for tabular self-supervised learning akin to SimSiam. Positive pairs are created by different regularizations in the forward pass. |
| Supervised | LogReg | Supervised training with logistic regression. |
| | Supervised | Supervised training with a classification layer added to the encoder used in other methods. |
| Ablation | Random Init | As an ablation baseline we report the accuracy using a randomly initialized encoder (Eldele et al., 2021). |

architecture (He et al., 2016). To avoid domain-specific projector design, in each case the random projectors reuse the encoder architecture, but are scaled down. Complete details are in Appendix C.

**Baseline methods:** Table 1 summarizes all baselines used in our experiments. It is worth noting that while our proposed framework LFR is domain-agnostic, popular SSRL methods such as SimCLR and SimSiam require domain-specific augmentations to achieve optimal performance. Specifically, the default augmentations used for view creation in SimCLR and SimSiam are designed for natural image classification, and may not be suitable for other modalities. In our experiments with tabular datasets, we compare our approach to SCARF (Bahri et al., 2022), which is a version of SimCLR adapted to tabular data that uses random corruptions as augmentations. For more detailed information on the implementations and augmentations, please refer to Appendix C.

## 4.3 PERFORMANCE ACROSS DATA MODALITIES

The performance of LFR and baselines across multiple modalities is shown in Table 2. Our experiments show that for modalities where there are no standardized augmentation pipelines such as medical images, time series, and tabular data, LFR had the strongest performance among the SSRL methods, outperforming other self-supervised learning methods in most cases including the domain-agnostic ones such as DACL. For instance, on the HAR and Epilepsy datasets, LFR was the best performing method, beating the time-series specific self-supervised learning method TS-TCC. Similarly, for the Income and Theorem datasets, LFR outperformed the tabular data specific self-supervised learning baselines SCARF and STab. Although on the HEPMASS dataset LFR was not the best, it still performed well, comparable to the autoencoder and SCARF. Interestingly, for the Income dataset, LFR even outperformed supervised training. For time series and tabular data, augmentation-based methods like SimSiam tend to underperform. For example, SimSiam was worse than a randomly initialized encoder in HAR and Income.

This experimental result reflects our hypothesis – it is feasible to learn high-quality data representations across all modalities tested by predicting random data projections. LFR shows comparatively good performance on domains where semantic-preserving augmentations are difficult to create.

## 4.4 PERFORMANCE ON MEDICAL APPLICATIONS

To evaluate the performance of LFR on specific application domains, we conducted a performance comparison on two medical datasets: Kvasir for medical images and MIMIC-III for medical time series (see shaded columns in Table 2). Our findings show that LFR performed the best on Kvasir and was highly comparable to other well-performing baselines on MIMIC-III.

It is worth noting that the standard image augmentation pipeline used in other self-supervised methods is heavily tailored towards natural images and does not lead to superior performance on medical images (Kvasir).[1] Similarly, the standard augmentations proposed for time series do not capture the

---

[1] Some images in the Kvasir dataset have unnatural green annotation boxes in the bottom left corner which illustrate the configuration of the endoscope as it captured the image (Pogorelov et al., 2017), which may conflict with standard image augmentations. It would require domain knowledge to craft suitable augmentations.

Table 2: Performance comparison across various data modalities and application domains. Results of the best self-supervised learning methods are in bold. Shaded columns denote medical applications.

|  |  | Time series | | | Tabular | | | Image |
|---|---|---|---|---|---|---|---|---|
|  |  | HAR | Epilepsy | MIMIC-III | Income | Theorem | HEPMASS | Kvasir |
|  | Log Reg | $57.5_{\pm N/A}$ | $80.9_{\pm N/A}$ | $47.8_{\pm N/A}$ | $84.8_{\pm N/A}$ | $45.3_{\pm N/A}$ | $90.7_{\pm N/A}$ | - |
|  | Supervised | $96.0_{\pm 0.6}$ | $98.3_{\pm 0.1}$ | $48.8_{\pm 0.0}$ | $81.5_{\pm 0.2}$ | $53.8_{\pm 0.5}$ | $91.5_{\pm 0.0}$ | $83.2_{\pm 0.2}$ |
| Self-supervised | Random Init | $80.7_{\pm 2.3}$ | $89.1_{\pm 0.1}$ | $42.4_{\pm 1.1}$ | $83.1_{\pm 0.2}$ | $44.9_{\pm 0.8}$ | $84.3_{\pm 1.3}$ | $28.9_{\pm 5.7}$ |
|  | Autoencoder | $77.2_{\pm 0.7}$ | $90.8_{\pm 1.3}$ | $44.9_{\pm 0.5}$ | $85.0_{\pm 0.1}$ | $50.0_{\pm 0.4}$ | $\mathbf{90.7_{\pm 0.0}}$ | $72.4_{\pm 0.6}$ |
|  | DIET | $88.6_{\pm 1.3}$ | $96.8_{\pm 0.3}$ | $33.8_{\pm 5.2}$ | $82.2_{\pm 0.4}$ | $47.1_{\pm 0.5}$ | - | $71.3_{\pm 0.9}$ |
|  | SimSiam | $65.1_{\pm 0.8}$ | $97.4_{\pm 0.0}$ | $41.0_{\pm 1.9}$ | $79.2_{\pm 1.9}$ | $40.9_{\pm 0.9}$ | $85.3_{\pm 3.1}$ | $72.6_{\pm 1.4}$ |
|  | SimCLR | $87.8_{\pm 0.4}$ | $97.4_{\pm 0.2}$ | $44.1_{\pm 0.1}$ | - | - | - | $72.1_{\pm 0.3}$ |
|  | SCARF | - | - | - | $84.2_{\pm 0.1}$ | $48.5_{\pm 1.0}$ | $90.1_{\pm 0.1}$ | - |
|  | STab | - | - | - | $84.2_{\pm 0.3}$ | $50.7_{\pm 0.7}$ | $83.6_{\pm 1.7}$ | - |
|  | TS-TCC | $91.2_{\pm 0.8}$ | $97.6_{\pm 0.2}$ | $38.5_{\pm 1.3}$ | - | - | - | - |
|  | DACL | $90.7_{\pm 0.4}$ | $97.5_{\pm 1.5}$ | $40.9_{\pm 0.6}$ | $79.8_{\pm 0.7}$ | $47.6_{\pm 1.0}$ | $88.7_{\pm 0.8}$ | $72.0_{\pm 0.1}$ |
|  | LFR (Ours) | $\mathbf{93.1_{\pm 0.5}}$ | $\mathbf{97.9_{\pm 0.2}}$ | $\mathbf{46.6_{\pm 0.3}}$ | $\mathbf{85.2_{\pm 0.1}}$ | $\mathbf{51.6_{\pm 0.7}}$ | $90.1_{\pm 0.2}$ | $\mathbf{74.9_{\pm 0.6}}$ |

unique characteristics of online monitoring data. In contrast, LFR does not require augmentations, making it a promising approach for application-specific datasets and tasks. Overall, our results suggest that LFR is an effective method for learning meaningful representations, even for specific applications that would otherwise require domain knowledge.

## 4.5 IMPACT OF RANDOM DATA PROJECTOR DIVERSITY

In this section, we analyze the impact of the diversity of random data projectors on LFR using the Kvasir dataset. Projector diversity can be influenced at two stages: initialization and selection. For projector selection, we used the Determinantal Point Process illustrated in Section 3.3. Regarding projector initialization, we employ two techniques: Beta initialization and weight dropout.

**Beta initialization**: We drew inspiration from the Prewitt operator (Prewitt, 1970) and initialized the weights of 2D convolutional layers to create diverse targets that emphasize various edge features. To achieve this, we used a scaled version of the Beta distribution with parameters $\alpha = 0.5$ and $\beta = 0.5$ to initialize the convolutional layer weights to be close to -1 and 1.

**Weight dropout**: We utilized DropConnect (Wan et al., 2013) to initialize the target networks. This involved randomly setting a fraction of weights to zero with a corruption rate of 0.4. Unlike standard DropConnect used for network regularization, we froze the weights after initialization to enhance diversity in the target representations.

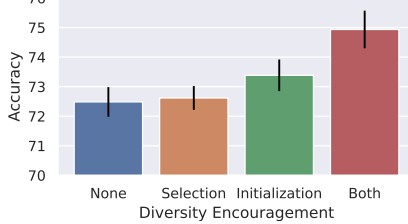

Figure 3: Effect of target diversity

**Results:** We compared the effectiveness of diversity encouragement at both the initialization and selection stages through an ablation on Kvasir. Our results, shown in Figure 3, demonstrate that promoting diversity in the projector set at both initialization and selection helps with the downstream performance. These findings underscore the importance of diversity in the projector set and highlight the effectiveness of our proposed methods for promoting diversity in LFR.

## 4.6 ABLATION STUDY

In this section, we investigate the impact of hyperparameters on the model's performance. To provide a more focused analysis of the trends observed in our ablation study, we present the results for Kvasir, which is representative of the other datasets. While we have observed similar trends in other datasets, the significance may vary depending on the specific characteristics of each.

**Number of projectors:** We evaluated the impact of the number of random projectors $K$ on linear evaluation accuracy using Kvasir data. We plotted the mean accuracy and standard deviation across

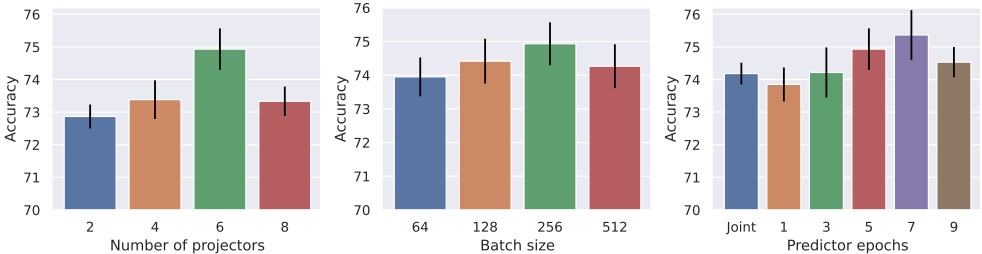

Figure 4: Test accuracy with different hyperparameters on Kvasir. **Left:** Number of random projectors. **Middle:** Batch size. **Right:** Predictor training setting.

3 runs for $K = 2, 4, 6$ and $8$ in Figure 4. The results indicate that the learned representation has a higher linear probing accuracy as the number of projectors increases until it reaches 6. Beyond $K = 6$, there is the possibility of reoccurring representations generated by the projectors, which can introduce bias into the encoder's training process. This could happen as the gradient descent optimization process might excessively favor redundant features, potentially leading to decreased performance. Based on these findings, we used $K = 6$ for our experiments on Kvasir.

**Batch size:** Because the selection of diverse projectors in LFR relies on one representative batch of data, we examine the sensitivity of LFR to training batch sizes. We plotted the linear accuracy with batch sizes of 64, 128, 256, and 512, and reported the mean accuracy and standard deviation across 3 runs in Figure 4. Our results indicate that LFR has relatively stable accuracies across batch sizes, with no significant difference between them. However, the best performing batch size on the Kvasir dataset was 256, and thus we used this batch size for our subsequent experiments.

**Predictor training epochs:** Another option in LFR is how the encoder $f_\theta$ and predictors $H_\Phi$ are trained. We tested joint training, where all models are updated together, and alternating training with various numbers of epochs for the predictors between each encoder epoch. From Figure 4, our findings on Kvasir indicate that updating the predictors for several epochs can improve performance compared to joint training, suggesting that more optimal predictors provide better learned representations.

**Embedding dimension:** We conducted an analysis on the effect of latent dimension on accuracy. Figure 5 demonstrates that as the latent dimension increases, there is a corresponding improvement in performance accuracy. However, when the latent dimension is more than 2048, the performance increase is minimal. Therefore, we used 2048 for our experiments.

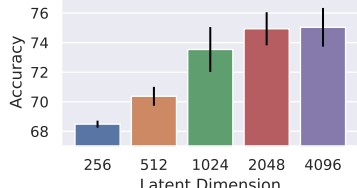

Additional experimental results are provided in Appendix D.

Figure 5: Test accuracy with different embedding dimensions.

## 5 CONCLUSION

This paper presents a novel self-supervised representation learning framework that is both modality-agnostic and application-agnostic. Our proposed framework utilizes random projectors to learn representations from unlabelled data, demonstrating excellent performance across various modalities and applications, particularly in situations where robust augmentation pipelines are not yet established.

The LFR technique is best suited to situations where the data cannot be reasonably augmented (due to the lack of domain knowledge). Although surprising, this situation occurs frequently in critical application domains, such as healthcare as we have highlighted. For general applications, however, if one knows the application domain well with adequate intuition around sensible data augmentations, using contrastive-learning-based SSRL is still likely to outperform random projectors. Overall, we treat LFR as a great complement to SSRL literature to fill the gap of data augmentation-free SSRL that satisfies the needs of many crucial applications.

While LFR encourages the use of SSRL across modalities and domains without the need for expert knowledge to craft augmentations, the limited human input to the learning process may increase the risk of sensitive features being misused, leading to privacy or fairness concerns. We recommend human oversight of self-supervised methods to monitor for appropriate use of data.

**Reproducibility Statement**    Towards the goal of reproducibility, we have provided our anonymized code repository as supplemantary material with this submission. The codebase includes instructions on how build the required environment, and how to run our proposed method as well as baseline methods. We also provide instructions for how to access the datasets, and run our pre-processing code. All relevant descriptions of the model architectures and hyperparameters are described in Section 4.2 and Appendix C. Additionally, an algorithmic description of our proposed method in pseudocode is provided in Appendix B. A statement on the computational resources spent is given in Appendix E. Additional notes on reproducibility are detailed in Appendix C.4.

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

# A  EXTENDED RELATED WORKS

Continuing our discussion of contrastive-learning based pretext tasks, we summarize the most common augmentation techniques used in contrastive learning and compare them across domains.

**Pretext tasks for contrastive learning**

Contrastive learning methods rely heavily on augmentation to generate positive views – semantically similar examples which are optimized to have the same representation as the original datapoint. To ensure the quality of learned representations, augmentations should be semantic-preserving (Tian et al., 2020; Balestriero et al., 2023). However, finding suitable augmentations for different application domains can be a challenging task, and researchers have invested considerable effort into this area to enhance downstream performance.

**Computer vision**   Image datasets benefit from a wide range of semantically similar augmentations, including random cropping and resizing, horizontal flipping, color jittering, converting to grayscale, and Gaussian blurring (He et al., 2020; Chen et al., 2020; Chen & He, 2021). However, the best performing augmentations are often dataset-specific (Ericsson et al., 2022) and require domain knowledge to determine. For instance, enforcing color invariance by grayscale conversion may not be beneficial for flower datasets (Zhang & Ma, 2022), but it can improve performance for ImageNet data (Chen et al., 2020). Furthermore, augmentations designed for natural images may not be suitable for medical domains (Elgendi et al., 2021). For instance, MoCo-CXR (Sowrirajan et al., 2021) focuses on chest X-ray images and uses random rotation and horizontal flipping instead of random crops, Gaussian blur, and random grayscale, as the latter may alter disease labels or are not meaningful for grayscale X-ray images. In another study, the authors proposed specific color space transforms for pathology images (Kang et al., 2023), as naïve color-jittering may produce unrealistic resulting images (Shen et al., 2022). Finally, the choice of augmentations can even be task-specific (Xiao et al., 2021). As an example, aggressive cropping may be suitable for image classification, but may not be optimal for image recognition tasks (Purushwalkam & Gupta, 2020).

**Time series**   Time series data often contains underlying patterns that are not easily identifiable by humans, unlike images with recognizable features (Luo et al., 2023). Consequently, designing effective data augmentation methods for time series data poses significant challenges and often requires domain knowledge. For example, augmentations for wearable sensor signals include rotation to simulate different sensor placements and jittering to simulate sensor noise (Um et al., 2017). Other researchers have focused on bio-signals and introduced channel augmentations that preserve the semantic information in the context (Mohsenvand et al., 2020; Cheng et al., 2020). Neighbourhood contrastive learning (Yèche et al., 2021) proposed leveraging patient identity information in online patient monitoring and using near-in-time data sequences of the same patient as semantically equivalent pairs. However, these augmentations are often specifically designed for the dataset and downstream task (Zhang & Ma, 2022), and their performance may deteriorate when applied to other time series data (Eldele et al., 2021). Therefore, identifying the optimal augmentation pipeline for each dataset and task requires extensive analysis (Iwana & Uchida, 2021).

**Tabular**   SSRL methods are understudied in the tabular domain as designing effective semantic-preserving augmentations is particularly challenging for structured data (Yoon et al., 2020). Like for time-series, it is often difficult for a human to determine if two views should be considered semantically equivalent, and unlike computer vision small changes to individual features can drastically change the content. SubTab (Ucar et al., 2021) proposed to generate positive views through different feature subsets. More recently, SCARF (Bahri et al., 2022) proposed to augment each record by corrupting a random subset of features. Finally, STab (Hajiramezanali et al., 2022) creates the contrastive views by imposing different regularization on the encoder for the same input.

# B  ALGORITHM

We summarize the LFR algorithm in Algorithm 1 which uses the subroutine in Algorithm 2.

---

**Algorithm 1** LFR: Learning From Randomness

---

**Require:** Dataset $\mathcal{D} = (x_i)_{i=1}^{n}$, number of random projectors $K$
**Ensure:** Encoder $f_\theta$
1: Initialize encoder $f_\theta$
2: Initialize $10 \cdot K$ different random projectors and select $K$ diverse projectors $g^1, ..., g^K$ using DPP as introduced in Section 3.3
3: Initialize $K$ predictors $h_\phi^1, ..., h_\phi^K$, one for each projector.
4: **for** epoch$_\text{all}$ in training epochs **do**
5:     **for** each mini-batch $B$ **do**                        ▷ Train encoder
6:         Train-Network($B$, $f_\theta$, $h_\phi^k$, $g^k$)
7:         Update parameters of $f_\theta$ with gradient descent using $L_{\text{BBT}}$ as introduced in Section 3.2
8:     **end for**
9:     **for** epoch$_p$ in predictor epochs **do**
10:         **for** each mini-batch $B$ **do**                ▷ Train predictor
11:             Train-Network($B$, $f_\theta$, $h_\phi^k$, $g^k$)
12:             Update parameters of all $h_\phi^k$ with gradient descent using $L_{\text{BBT}}$
13:         **end for**
14:     **end for**
15: **end for**

---

**Algorithm 2** Train-Network subroutine

---

1: **procedure** TRAIN-NETWORK($B$, $f_\theta$, $h_\phi^k$, $g^k$)
2:     Compute representations: $Z = f_\theta(B)$
3:     **for** $k = 1$ to $K$ **do**
4:         Compute output representations: $h_\phi^k(Z)$
5:         Compute representation from projector $k$: $g^k(Z)$
6:         Compute loss $L_{\text{BBT}}$
7:     **end for**
8: **end procedure**

---

# C  IMPLEMENTATION DETAILS

## C.1  DATASET SUMMARY

Table 3 provides a summary of all datasets used in our experiments, along with the corresponding downstream tasks and evaluation metrics.

Table 3: Dataset description.

| Dataset | Modality | Data Domain | Train Size | Test Size | Downstream Task | Metric |
|---|---|---|---|---|---|---|
| HAR | Time series | Mobile sensors | 7352 | 2947 | Multi-class classification (6) | Accuracy |
| Epilepsy | Time series | Brain EEG | 9200 | 2300 | Binary classification | Accuracy |
| MIMIC-III | Time series | Patient Online Monitoring | 2,568,619 | 563,742 | Multi-class classification (10) | Cohen's Kappa |
| Fault Diagnosis (App. D.1) | Time series | Motor current signals | 8184 | 2728 | Multi-class classification (3) | Accuracy |
| Income | Tabular | Census | 30162 | 15060 | Binary classification | Accuracy |
| Theorem | Tabular | Logic Reasoning | 3059 | 1530 | Multi-class classification (6) | Accuracy |
| HEPMASS | Tabular | Particle Physics | 7,000,000 | 3,500,000 | Multi-class classification (2) | Accuracy |
| Kvasir | Image | Medical Images | 6000 | 2000 | Multi-class classification | Accuracy |
| CIFAR10 (App. D.6) | Image | Natural Images | 50000 | 10000 | Multi-class classification (10) | Accuracy |

Table 4: Details on LFR architectural parameters

| Dataset | Projectors | Projector Initialization | Encoder Architecture | Projector Architecture |
|---|---|---|---|---|
| HAR/Epilepsy/Fault Diagnosis (App D.1) | 6 | Pytorch Default | Three-block CNN | Two-block CNN |
| Income/Theorem | 6 | Pytorch Default | Four-layer MLP | Two-layer MLP |
| HEPMASS | 6 | Pytorch Default | Four-layer MLP | Two-layer MLP |
| MIMIC-III | 10 | Pytorch Default | Five-block TCN | Three-block TCN |
| Kvasir | 6 | $\beta$ Initializtion + Weight Dropout | ResNet18 | Four-layer CNN |
| CIFAR10 (App D.6) | 40 | $\beta$ Initializtion + Weight Dropout | ResNet18 | Four-layer CNN |

## C.2 NEURAL NETWORK ARCHITECTURES

**HAR/Epilepsy/Fault Diagnosis:** For LFR, we used a three-block convolutional network from (Wang et al., 2017; Eldele et al., 2021) as the representation model $f_\theta$. For the predictors $h_\phi^{(k)}$, we used a single linear layer. For the random projectors $g^{(k)}$, we adopted a similar architecture to the representation model but with slightly decreased complexity, a two-block convolutional network with 16 and 32 channels, followed by two sequential linear layers with a hidden dimension of 256. For all other self-supervised methods, we used the same representation model for a fair comparison. For SimCLR (Chen et al., 2020) and SimSiam (Chen et al., 2020), we used the same predictors as LFR, and a 3-layer ReLU network of hidden dimension 512 as a projector. The first two linear layers are followed by a batchnorm layer. To create the contrastive view for SimCLR (Chen et al., 2020) and SimSiam (Chen et al., 2020), we adopted the same augmentations as designed in TS-TCC (Eldele et al., 2021).

**MIMIC-III:** For all methods, we followed the encoder structure from (Yèche et al., 2021) as the representaion model/encoder, with the exception that we used flattened temporal convolutional network (TCN) features followed by a linear layer, which produced the embedding size of 64. We also disabled the L2 normalization in the encoder. For the random projectors in LFR, we adopted a three-block TCN with kernel size of 2, followed by a linear layer with output channel size of 64 for each layer. The two-layer ReLU predictor is shared in LFR, SimCLR and SimSiam with a hidden dimension of 256. We used the same projector and augmentation as in HAR/Epilepsy for SimCLR and SimSiam.

**Income/Theorem:** For LFR, we followed the setup in (Hajiramezanali et al., 2022; Bahri et al., 2022) and used a 4-layer ReLU network with a hidden dimension of 256 as the representation model, with a single linear layer predictor. The random projector networks had a similar architecture but were less complex, using a 2-layer ReLU network with a hidden dimension of 256. For the contrastive baselines, we employed the same encoder and predictor for a fair comparison, and followed (Bahri et al., 2022) by using a 2-layer ReLU network with a hidden dimension of 256 as projectors. To generate the contrastive views, we used the SCARF (Bahri et al., 2022) augmentation technique to randomly corrupt features with values sampled from their empirical distribution, ensuring that our SimCLR baseline was identical to SCARF.

**HEPMASS:** For the HEPMASS dataset, we used the same network architecture as for the Income/Theorem datasets but with the output latent dimension of the encoder set to 16.

**Kvasir/CIFAR10:** For LFR, we used ResNet18 (He et al., 2016) as the representation model, with an output dimension of 2048 for all datasets. For CIFAR10 we followed SimSiam (Chen et al., 2020) to use the CIFAR variant of ResNet18. The predictors are 4-layer ReLU networks with a hidden dimension of 256. For the random projector networks, we adopted a 4-layer CNN of channels $[3, 8, 16, 32]$, each layer is followed by a ReLU, and every two layers are followed by max-pooling. A linear layer is used to map the output to 2048 dimensions. For the contrastive baselines, we adopted the same representation model for a fair comparison and used 3-layer ReLU networks with hidden dimension 256 as projectors. To create the contrastive views, we employed the augmentation set adopted by SimSiam Chen et al. (2020) for CIFAR (`RandomResizedCrop`, `RandomHorizontalFlip`, `ColorJitter`, `RandomGrayscale`) and added `GaussianBlur` for Kvasir. We applied the same set of augmentations to the implementation of DIET (Balestriero, 2023). All other settings are the same as with the ResNet18 model.

All supervised baselines use the same representation model as the SSL methods, with the final layer being a linear classification layer.

We summarize all architectural related settings of LFR in Table 4

## C.3 DETAILS OF TRAINING SETTINGS

**LFR training settings:** Table 5 summarizes all the training settings used for LFR, while Table 6 outlines the evaluation settings used for downstream tasks. We used a Logistic Regression classifier for all tabular datasets including Income, Theorem, and HEPMASS, while for MIMIC-III we used a MLP network to predict the length of stay following (Yèche et al., 2021). For the remaining datasets, we followed prior works such as TS-TCC (Eldele et al., 2021), SimSiam (Chen et al., 2020), and BYOL (Grill et al., 2020) by using a linear classifier for classification tasks.

Table 5: Details on LFR Training Settings

| Dataset | Optimizer | Batch Size | Learning Rate | Optimizer Parameters | Epochs |
|---|---|---|---|---|---|
| HAR/Epilepsy/Fault Diagnosis (App D.1) | Adam | 128 | 3e-4 | $\beta$=(0.9, 0.999), wd=3e-4 | Train epochs = 200, Predictor epochs = 5 |
| Income/Theorem | Adam | 128 | 1e-3 | $\beta$=(0.9, 0.999), wd=0 | Train epochs = 100, Predictor epochs = 1 |
| HEPMASS | Adam | 512 | 1e-6 | $\beta$=(0.9, 0.999), wd=0 | Train epochs = 20, Predictor epochs = 1 |
| MIMIC-III | Adam | 4096 | 1e-3 | $\beta$=(0.9, 0.999), wd=5e-4 | Train steps = 600, Predictor steps = 5 |
| Kvasir | SGD | 256 | 1e-4, cosine decay | momentum=0.9, wd=5e-4 | Train epochs = 400, Predictor epochs = 5 |
| CIFAR10 (App D.6) | SGD | 512 | 3e-2, cosine decay | momentum=0.9, wd=5e-4 | Train epochs = 400, Predictor epochs = 5 |

Table 6: Details on Linear Evaluation Settings of SSL methods

| Dataset | Optimizer | Batch size | Learning Rate | Optimizer Parameters | Epochs |
|---|---|---|---|---|---|
| HAR/Epilepsy | Adam | 128 | 3e-4 | $\beta$=(0.9, 0.999), wd=3e-4 | 100 |
| MIMIC-III | Adam | 4096 | 1e-4 | $\beta$=(0.9, 0.999), wd=5e-4 | 300 |
| Kvasir | SGD | 256 | 1e-3, cosine decay | momentum=0.9, wd=0 | 100 |
| CIFAR10 (App D.6) | SGD | 512 | 0.2, cosine decay | momentum=0.9, wd=0 | 100 |

**Baseline training settings:** All self-supervised baselines adopt the same training setting as LFR unless stated otherwise. For DIET with CIFAR10, we followed the original paper to train 5000 epochs with 10 linear warmup epochs and then cosine decay. For DIET with MIMIC-III, we used batch size 512 and trained for 2000 steps with 10 warmup epochs. We reserved 5000 epochs for training the autoencoder on MIMIC-III with 500 warmup epochs. For other self-supervised methods with MIMIC-III, we also added 60 warmup epochs. We summarize the training settings of supervised baselines in Table 7.

Table 7: Details on Supervised Training Settings

| Dataset | Optimizer | Batch size | Learning Rate | Optimizer Parameters | Epochs | Augmentations |
|---|---|---|---|---|---|---|
| HAR/Epilepsy/Fault Diagnosis (App D.1) | Adam | 128 | 3e-4 | $\beta$=(0.9, 0.999), wd=3e-4 | 500 | None |
| Income/Theorem | Adam | 128 | 1e-3 | $\beta$=(0.9, 0.999), wd=0 | 100 | None |
| MIMIC-III | Adam | 4096 | 5e-6 | $\beta$=(0.9, 0.999), wd=5e-4 | 10 | Same as SimCLR and SimSiam |
| Kvasir | SGD | 256 | 1e-2, cosine decay | momentum=0.9, wd=0 | 600 | Same as SimCLR and SimSiam |
| CIFAR10 (App D.6) | SGD | 512 | 3e-2, cosine decay | momentum=0.9, wd=5e-4 | 800 | Same as SimCLR and SimSiam |

## C.4 REPRODUCIBILITY NOTES

**TS-TCC:** Our results for TS-TCC on the HAR and Epilepsy datasets had several discrepancies with the values reported in the original TS-TCC paper (Eldele et al., 2021). We discovered that in the official implementation of TS-TCC, the input data was augmented once and then kept the same throughout training, rather than being randomly augmented in each forward pass. We fixed this bug and were able to achieve *better* results. Additionally, we increased the number of training epochs for our supervised baseline, which also led to improved performance. Lastly, we noticed that in the original TS-TCC implementation, the random initialization ablation was evaluated using a randomly initialized linear classification head that was not trained, whereas we evaluated with a trained linear classification layer and saw a significant increase in accuracy for this ablation.

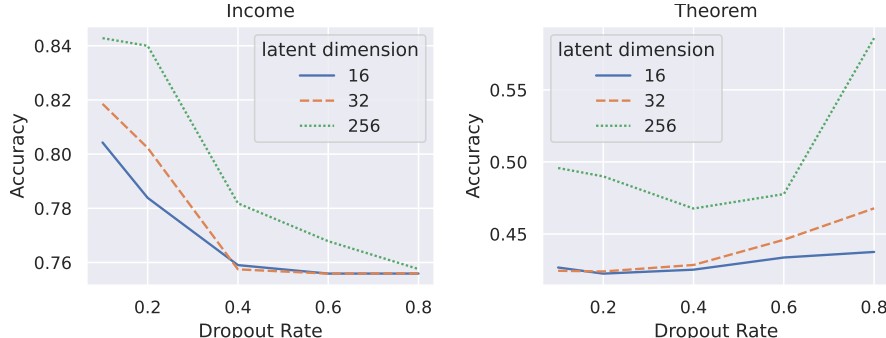

Figure 6: STab accuracy as a function of dropout rate

**STab:** The original STab paper (Hajiramezanali et al., 2022) did not provide information about the random DropConnect ratio or training hyperparameters used in their experiments. In our implementation, we used the same training hyperparameters as other SSL methods and tested DropConnect ratios of 0.1, 0.2, 0.4, 0.6, and 0.8, with the results shown in Figure 6. We selected the best-performing ratio for each experiment and reported the corresponding results. We ended up selecting 0.1 for Income and 0.8 for Theorem.

## D ADDITIONAL EXPERIMENTAL RESULTS

### D.1 TRANSFER LEARNING EXPERIMENTS

We follow the settings in prior work (Eldele et al., 2021) and perform transfer learning experiments on a set of four real-world Fault Diagnosis datasets (Ragab et al., 2020). Each dataset was collected under a different working condition and can be considered as a separate domain with different characteristics. As shown in Table 8, LFR is able to learn features on the source task that transfer well to related tasks, outperforming the supervised approach (with the same model architecture) on 10 out of 12 dataset pairs, TS-TCC on 8 pairs, and is significantly better than all other approaches in terms of average performance. Hence, we see that LFR is able to transfer representations in a tabular setting, at least when there is some similarity in the domain of the datasets used.

Table 8: Transfer learning experiments

| Method (Encoder Arch.) | A→B | A→C | A→D | B→A | B→C | B→D | C→A | C→B | C→D | D→A | D→B | D→C | AVG |
|---|---|---|---|---|---|---|---|---|---|---|---|---|---|
| Supervised (Transf.) | 34.28 | 44.94 | 34.57 | 52.93 | 63.67 | 99.82 | 52.93 | 84.02 | 83.54 | 53.15 | 99.56 | 62.43 | 63.83 |
| Supervised (CNN) | 42.96 | 46.33 | 46.99 | 43.55 | 70.53 | 94.94 | 48.50 | 78.34 | 74.34 | 52.71 | 99.05 | 70.20 | 64.04 |
| Rand Init (CNN) | 79.55 | 68.80 | 79.95 | 78.96 | 58.39 | 81.34 | 70.60 | 84.93 | 80.02 | 80.10 | 80.13 | 57.77 | 75.05 |
| TS-TCC (Transf.) | 43.15 | 51.50 | 42.74 | 47.89 | 70.38 | 99.30 | 38.89 | 98.31 | 99.38 | 51.91 | 99.96 | 70.31 | 67.83 |
| LFR (CNN) | 90.51 | 81.63 | 93.40 | 75.99 | 75.48 | 88.64 | 69.32 | 80.54 | 88.34 | 78.92 | 87.90 | 75.26 | 82.16 |

### D.2 EMBEDDING DIMENSIONS

As we discussed in Section 4.6, the dimensionality of embeddings may have a strong effect on the richness of learned representations. To complement the image-based results on Kvasir presented in the main text, we also used the Theorem dataset to evaluate the performance of LFR and baseline SSRL approaches across latent dimension sizes. Figure 7 shows that increasing the latent dimension improved the accuracy of each approach up to about 256. LFR consistently outperformed all the other baselines across all the latent dimension settings.

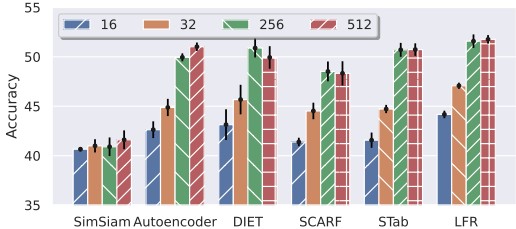

Figure 7: Effect of embedding dimension on LFR and self-supervised learning baselines with the Theorem dataset.

Table 9: Effect of sample size during random projector selection

| Sample size | 64 | 128 | 256 | 512 |
|---|---|---|---|---|
| Accuracy | $74.1 \pm 0.4$ | $74.0 \pm 0.4$ | $74.9 \pm 0.6$ | $74.1 \pm 0.4$ |

### D.3 SAMPLE SIZE FOR PROJECTOR SELECTION

As illustrated in Section 4.5, we use a sufficiently large set of data to select a diverse set of random projectors. In this section, we investigated the effect of the sample size for random projector selection with the Kvasir dataset. As shown in Table 9, the sample size does not heavily affect the accuracy (the fluctuation is less than 1.0%). Therefore, for the ease of implementation, we use the same size of the batch size in training, i.e. 256 for Kvasir.

### D.4 FINE-TUNING PERFORMANCE

In Section 4 we evaluated SSRL methods on downstream tasks using linear evaluation. SSRL methods are sometimes finetuned to a downstream task, so below we compare LFR's finetuning performance to benchmark methods.

During the finetuning phase, we employed a linear layer combined with labeled data to enhance the performance of the encoder. The architecture setup was consistent with that detailed in the main paper. However, recognizing the significant volume of data in larger datasets such as MIMIC-III and HEPMASS, we chose a semi-supervised learning approach in those cases. In this strategy, we randomly selected 10% of labeled data from these datasets to facilitate the fine-tuning process. This decision was driven by the computational resources required when dealing with extensive data. Conversely, for smaller datasets, we conducted fine-tuning using the complete set of available labeled data, enabling us to evaluate the model's performance across the entirety of the datasets.

As shown in Table 10, through the fine-tuning process all methods exhibit more comparable performance across the datasets. LFR still achieved the best performance on a majority of the datasets we used, although with overlapping error bars to other methods in those cases.

Table 10: Performance comparison among the SSRL methods with finetuning.

| | Time series | | | Tabular | | | Image |
|---|---|---|---|---|---|---|---|
| | HAR | Epilepsy | MIMIC-III | Income | Theorem | HEPMASS | Kvasir |
| Supervised | $96.0 \pm 0.6$ | $98.3 \pm 0.1$ | $48.8 \pm 0.0$ | $81.5 \pm 0.2$ | $53.8 \pm 0.5$ | $91.5 \pm 0.0$ | $83.2 \pm 0.2$ |
| Autoencoder | $93.9 \pm 1.3$ | $95.1 \pm 2.0$ | $49.2 \pm 0.6$ | $85.2 \pm 0.1$ | $53.9 \pm 0.5$ | $90.8 \pm 0.0$ | $75.0 \pm 0.8$ |
| DIET | $\mathbf{95.6} \pm \mathbf{0.5}$ | $97.8 \pm 0.1$ | $48.4 \pm 0.1$ | $85.2 \pm 0.1$ | $52.4 \pm 0.9$ | - | $74.4 \pm 0.3$ |
| SimSiam | $93.4 \pm 0.6$ | $97.9 \pm 0.2$ | $49.4 \pm 0.3$ | $85.2 \pm 0.1$ | $52.5 \pm 0.8$ | $90.7 \pm 0.0$ | $74.5 \pm 0.6$ |
| SimCLR | $93.7 \pm 1.1$ | $97.8 \pm 0.2$ | $48.6 \pm 0.8$ | - | - | - | $74.5 \pm 0.6$ |
| SCARF | - | - | - | $85.1 \pm 0.2$ | $53.8 \pm 0.8$ | $90.9 \pm 0.0$ | - |
| STab | - | - | - | $85.3 \pm 0.2$ | $53.0 \pm 0.7$ | $\mathbf{91.1} \pm \mathbf{0.0}$ | - |
| LFR | $94.7 \pm 1.4$ | $\mathbf{98.2} \pm \mathbf{0.2}$ | $\mathbf{49.6} \pm \mathbf{0.1}$ | $85.3 \pm 0.1$ | $\mathbf{54.3} \pm \mathbf{0.4}$ | $90.8 \pm 0.0$ | $\mathbf{75.5} \pm \mathbf{0.7}$ |

### D.5 LOSS FUNCTION ABLATIONS

In the main paper, we used the batch-wise Barlow Twins loss as a divergence measure, however this specific choice of loss function is not required for our method, and many other choices of contrastive loss could also work. As an ablation, we also ran LFR with the InfoNCE loss on the Kvasir dataset and observed a slight decrease compared to the Barlow Twins loss ($72.6 \pm 0.4$ compared with $74.9 \pm 0.6$).

### D.6 PERFORMANCE ON CIFAR10

The primary focus of LFR is domain-agnostic representation learning, mainly in cases where it is not clear how to usefully augment the data. As a result, we focused on tabular, time-series, and medical

Table 11: Performance on CIFAR10

| Method | Supervised | Random Init | Autoencoder | DIET | SimSiam | SimCLR | DACL | LFR |
|---|---|---|---|---|---|---|---|---|
| Accuracy | $94.4 \pm 0.2$ | $34.7 \pm 0.3$ | $37.4 \pm 0.3$ | $69.3 \pm 1.2$ | $89.2 \pm 0.1$ | $86.7 \pm 0.5$ | $42.4 \pm 2.7$ | $64.3 \pm 0.1$ |

imaging applications in the main paper. Here, we explore the effectiveness of LFR on natural images with CIFAR10 data. As shown in Table 11, as one might expect, SimSiam and SimCLR achieve the best performance out of our baselines thanks to their extensively engineered pipeline of image augmentations. Two domain-agnostic approaches that do not use image augmentations, DIET and our LFR, fall into a second tier, outpacing the third tier of DACL, Autoencoder, and Random Init. This result is in stark contrast to what we found on Kvasir, where all methods ended up in a tight range, with LFR at the top. This suggests that the heavy augmentations of SimSiam and SimCLR really are specialized for natural images, and are not suitable for other domains within the image modality.

### D.7 RANDOM PROJECTORS VISUALIZATION

To illustrate the behaviour of individual random projections, and the role of diversity selection, we trained LFR on a small image-based dataset, CIFAR10[2], and examined the nearest-neighbour representations for different projectors. First we selected a query image at random, then for each of 100 randomly initialized projectors, we encoded all training images, and found the nearest encoded representations to the query. Distances in embedding space are measured by cosine similarity. We then visually compared the nearest neighbours that were selected by pairs of projectors which were deemed most diverse or similar according to the DPP criterion from Equation (9). The pairs with the highest diversity (Diverse 1 and Diverse 2) and the highest similarity (Similar 1 and Similar 2) are shown in Figure 8.

The results in Figure 8 show that our similarity measure is effective at selecting projectors that focus on diverse features. For example, in Figure 8 on the frog query (top left) we see that the projector in the first row represented a white "edge" feature, since the nearest neighbours all share that feature from the query but are otherwise semantically different. The projector in the second row appears to focus on shape, and its nearest neighbours are very different from the first projector. On the other hand, looking at the bottom right set of images, the most similar projector pair selects very similar nearest neighbors with 2 out of 5 nearest neighbors being identical. This qualitative study suggests that promoting diversity in the projectors can lead to the capture of a broader range of features. Then, diverse features will be available to the representation model during training, potentially resulting in a richer set of semantic knowledge and ultimately better performance on downstream tasks. This interpretation is supported by the evidence in Section 4.6 that encouraging projector diversity results in better performance on linear evaluation accuracy.

## E COMPUTATIONAL RESOURCES AND TIME SPENT

The time series experiments with HAR and Epilepsy were conducted on a Tesla V100 GPU with 32 GB of memory, except for TS-TCC which was conducted on a TITAN V with 12 GB of memory. The experiments took a total of 102 GPU hours, including all baseline experiments. The MIMIC-III experiments were conducted with an NVIDIA A100 GPU with 40GB of memory, except for TS-TCC which was again conducted on a TITAN V with 12 GB of memory, and cost 608 GPU hours, including all baseline experiments. The Kvasir experiments were conducted using a Tesla V100 GPU with 32 GB of memory, and they took a total of 1095 GPU hours, including all baseline experiments. The tabular dataset experiments with Income, Theorem, and HEPMASS were conducted on an NVIDIA TITAN V GPU with 12 GB of memory. The experiments took a total of 70 GPU hours, including all baseline experiments. The CIFAR experiments were conducted on a cluster with a single NVIDIA P100 GPU with 12 GB of memory per experiment, and they took a total of 3020 GPU hours, including all baseline experiments.

---

[2]CIFAR10 was chosen for this qualitative study rather than the other datasets we used because it has human-interpretable features and visualizations.

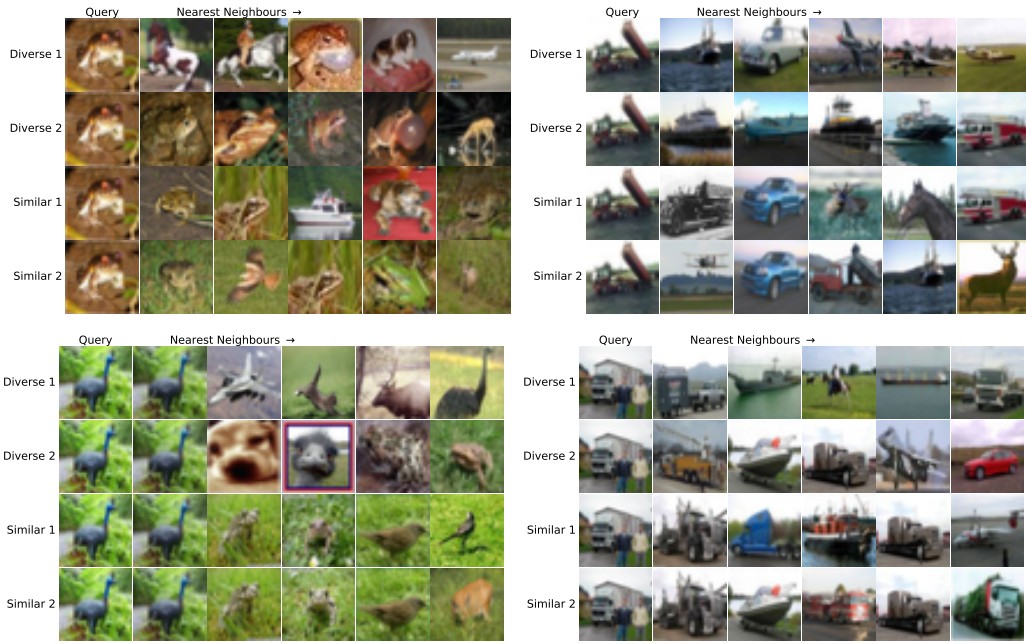

Figure 8: Nearest neighbors identified by randomly initialized projector models. Each row shows the query image (left) and its five nearest neighbours identified by a projector. The top two rows display the most dissimilar projectors, while the bottom two rows give the most similar projectors.

## F    NOTE ON COMPUTATIONAL EFFICIENCY OF LFR

Our experiments across diverse datasets consistently reveal that LFR's training time is often comparable to, or even notably shorter than the contrastive SSRL methods. For example, on Kvasir training SimCLR took around 9 hours, while LFR completed in just 2 hours on the same machine. This assessment of training time considers both CPU and GPU usage.

The efficiency in LFR's training time can be attributed to two primary factors. First, existing SSRL methods rely on extensive augmentations that are performed on CPU and can lead to data loading bottlenecks. LFR has no such issue. CPU-based image augmentations account for a significant amount of SimCLR's 9 hour training time on Kvasir. For datasets with less computationally intensive augmentations, training times tend to align more closely - on the HAR dataset, LFR concluded training in approximately 3 hours while SCARF required around 3.3 hours.

Second, multiple passes through predictor heads in LFR is not overly burdensome. Unlike contrastive frameworks that use double augmentation and encoder passes, LFR requires fewer expensive operations: one pass through each random projector before training, and one pass through the encoder followed by one pass through each small predictor during training. Given that the encoder is the largest network by far, LFR's training time often proves less than SimCLR which uses multiple encoder passes.

Compared to the cost of training an encoder, the diversity selection procedure using the Fast Determinantal Point Process (Chen et al. 2018) is an almost negligible contribution to the total training time. To recap, to select $K$ diverse projectors we initialize $N$ small neural networks and for each we run one forward pass on a batch of training data of size $m$. Then, DPP is applied over matrices of size $m^2 \times K$ constructed from the network outputs. As a concrete example, on the Income dataset training LFR for 100 epochs took 849 s on our machine, while the DPP selection took 3 s.

## G    MATHEMATICAL DERIVATION

Here we provide more details on the derivation of Equations 2 through 4.

Consider the MLE objective in Equation 2. As the variable $\mathbf{z}_i$'s distribution could be intractable, we represent it in a general form by the distribution $q(\mathbf{z}_i)$ such that

$$\sum_i \sum_k \log \int_{\mathbf{z}_i} p\left(\mathbf{y}_i^{(k)} \mid \mathbf{z}_i\right) p\left(\mathbf{z}_i \mid \mathbf{x}_i\right) d\mathbf{z}_i = \sum_i \sum_k \log \int_{\mathbf{z}_i} q(\mathbf{z}_i) \frac{p\left(\mathbf{y}_i^{(k)} \mid \mathbf{z}_i\right) p\left(\mathbf{z}_i \mid \mathbf{x}_i\right)}{q(\mathbf{z}_i)} d\mathbf{z}_i.$$

Using Jensen's inequality, the lower bound of the objective is therefore

$$\sum_i \sum_k \log \int_{\mathbf{z}_i} q(\mathbf{z}_i) \frac{p\left(\mathbf{y}_i^{(k)} \mid \mathbf{z}_i\right) p\left(\mathbf{z}_i \mid \mathbf{x}_i\right)}{q(\mathbf{z}_i)} d\mathbf{z}_i \geq \sum_i \sum_k \int_{\mathbf{z}_i} q(\mathbf{z}_i) \log \frac{p\left(\mathbf{y}_i^{(k)} \mid \mathbf{z}_i\right) p\left(\mathbf{z}_i \mid \mathbf{x}_i\right)}{q(\mathbf{z}_i)} d\mathbf{z}_i.$$

Hence, to maximize the variational lower-bound of the above equation with respect to the proposed distribution $q(\mathbf{z}_i)$, the optimal solution is simply to let equality hold (as in the classic EM algorithm),

$$\sum_i \sum_k \log \int_{\mathbf{z}_i} q(\mathbf{z}_i) \frac{p\left(\mathbf{y}_i^{(k)} \mid \mathbf{z}_i\right) p\left(\mathbf{z}_i \mid \mathbf{x}_i\right)}{q(\mathbf{z}_i)} d\mathbf{z}_i = \sum_i \sum_k \int_{\mathbf{z}_i} q(\mathbf{z}_i) \log \frac{p\left(\mathbf{y}_i^{(k)} \mid \mathbf{z}_i\right) p\left(\mathbf{z}_i \mid \mathbf{x}_i\right)}{q(\mathbf{z}_i)} d\mathbf{z}_i,$$

which, in turn, requires the following equation to hold,

$$\frac{p\left(\mathbf{y}_i^{(k)} \mid \mathbf{z}_i\right) p\left(\mathbf{z}_i \mid \mathbf{x}_i\right)}{q(\mathbf{z}_i)} = C,$$

where $C$ is a constant. Thus, the optimal solution for $q(\mathbf{z}_i)$ is

$$q(\mathbf{z}_i) = \frac{p\left(\mathbf{y}_i^{(k)} \mid \mathbf{z}_i\right)}{p\left(\mathbf{y}_i^{(k)} \mid \mathbf{x}_i\right)} p\left(\mathbf{z}_i \mid \mathbf{x}_i\right).$$

As both $p\big(\mathbf{y}_i^{(k)} \mid \mathbf{z}_i\big)$ and $p\big(\mathbf{y}_i^{(k)} \mid \mathbf{x}_i\big)$ are delta distributions with probability 1 conditioned on deterministic functions modelled by $g^{(k)}$ and $h_\phi^{(k)}$ respectively, the optimal solution of $q(\mathbf{z}_i)$ is simply $p(\mathbf{z}_i \mid \mathbf{x}_i)$ given the conditions are satisfied. In other words, we need $p(\mathbf{z}_i \mid \mathbf{x}_i)$ that can let $\mathbf{y}_i^{(k)} = h_\phi^{(k)}(\mathbf{z}_i)$ for all $k$. Thus, the optimization is essentially an EM algorithm where we optimize $\theta$ and $\phi$ alternatively to gradually increase the conditional likelihood of Equation 2.

