# OpenReview forum: "Self-supervised Representation Learning from Random Data Projectors"
_ICLR.cc/2024/Conference — ICLR 2024 poster_

### Official Review · Reviewer_sQMY · 2023-10-25

**Soundness:** 2 fair
**Presentation:** 3 good
**Contribution:** 2 fair
**Rating:** 6
**Confidence:** 4

**Summary:**

This work introduces a self-supervised representation learning scheme that can be applied to any data modality and network architectures. To this end, it proposed to learn the representations from random data projects of the input data. It is a scheme that learns from randomness, aiming to extract meaningful representations from randomness that mimic arbitrary downstream tasks.

**Strengths:**

1, this framework can accommodate various data modality

2, extensive experimental results with good performance

**Weaknesses:**

1, Despite the good performance of the experiments, what has been learned from the latent space using the scheme introduced in this paper, even intuitively?

2, It says in the paper that $g^{(k)}(x)$ uses the same architecture design of $f_{\theta}$. Even with random initialization, it may still follow a certain distribution family. How would changing this architecture affect the learning?

3, The paper only exams the accuracy on the classification task for frozen representations. Nonetheless, a good representation could used for various purposes, i.e., manipulation of each dimension in the latent space for generating new data, understanding the essential dynamics of the system in physical models and time series data. How could this strategy be applied to scenarios beyond classification?

4, Although the fact that choosing good augmentations usually requires domain knowledge, in many of the application mentioned, such knowledge could be partially obtained using physical intuition. By comparing with models under random corruptions doesn't seem to be a fair comparison.

**Questions:**

Please see Weaknesses.

---

> ### Author Response · Authors · 2023-11-18
> **Initial Response**
>
> Thank you for your review and comments about our method’s good performance. We can discuss each of your points.
>
> **W1: Despite the good performance of the experiments, what has been learned from the latent space using the scheme introduced in this paper, even intuitively?**
>
> While we do not have a complete theoretical explanation for the success of learning from randomness, we can offer you some indication of an interesting direction. Random orthogonal projections on high dimensional spaces have been studied in mathematics and found to have interesting properties. One result, known as the Johnson-Lindenstrauss lemma [A], says that a set of points on a high-dimensional Euclidean space can be embedded into a lower-dimensional space such that distances are preserved. It has been observed that distances are still well-preserved when the embedding is a random orthogonal projection. This rigorous mathematical setting loosely mimics the practical setting of our paper - where high-dimensional data is projected into a low-dimensional space via a randomly initialized network. While the network does not produce an orthogonal embedding, it may still approximately preserve the distance between datapoints. This information can then be leveraged by the representation learner. We anticipate that the research community will be interested in exploring more theoretical explanations after publication.
>
> **W2: It says in the paper that g^(k)(x) uses the same architecture design of f_theta. Even with random initialization, it may still follow a certain distribution family. How would changing this architecture affect the learning?**
>
> There is literature (e.g. [B]) discussing what distributions are produced by randomly initialized networks, so we certainly agree with your point. We did not have space or capacity to give a full analysis of the types of distributions produced by each of the networks we used, but we hope that members of the community will find that direction interesting to pursue.
>
> We found it sensible to use random projectors that share a similar architecture to the representation model since researchers have already determined classes of network architectures that have good inductive biases for extracting useful features. This information dictates the choice of representation model, and we reuse that information in choosing the projector architecture. In principle one could use any projector, but in preliminary experiments we found that matching inductive biases was beneficial.
>
> **W3: The paper only exams the accuracy on the classification task for frozen representations. Nonetheless, a good representation could used for various purposes, i.e., manipulation of each dimension in the latent space for generating new data, understanding the essential dynamics of the system in physical models and time series data. How could this strategy be applied to scenarios beyond classification?**
>
> As with any SSRL model, training is done without reference to any specific downstream task. Evaluation is commonly performed on classification tasks since class labels are widely available, but it would be possible to predict any kind of label, such as segmentations or depth in image data. On tabular datasets which were our main focus, there are not many examples of downstream tasks other than classification for which standardized datasets are publicly available. However, please see our general comment for a new experiment on transfer learning.
>
> **W4: Although the fact that choosing good augmentations usually requires domain knowledge, in many of the application mentioned, such knowledge could be partially obtained using physical intuition. By comparing with models under random corruptions doesn't seem to be a fair comparison.**
>
> We agree that with time and effort, it would be possible to produce useful semantic-preserving augmentations for most datasets, however our focus is on circumventing that process entirely. The advantage of LFR is that one does not need to put any effort into choosing good augmentations. In your last sentence we believe you are referring to one of the baseline methods, SCARF (Bahri et al., 2022), reviewed in Section 4.2. We believe that comparing to SCARF as one of the baselines is fair, since the random corruptions are exactly what (Bahri et al., 2022) proposed to use. We note that this is the only baseline that uses random corruptions. DACL for instance uses mixup for augmentations, while STab uses stochastic regularizations of networks.
>
> [A] Freksen, “An Introduction to Johnson-Lindenstrauss Transforms” 2021
>
> [B] Roberts et al., “The Principles of Deep Learning Theory”, Cambridge University Press 2022

---

> > ### Comment · Reviewer_sQMY · 2023-11-21
> > **Response**
> >
> > I want to express my gratitude to the reviewers for addressing my inquiries and to the other individuals providing reviews. As a result, I am increasing my rating.

---

### Official Review · Reviewer_Eryk · 2023-10-30

**Soundness:** 3 good
**Presentation:** 4 excellent
**Contribution:** 3 good
**Rating:** 6
**Confidence:** 5

**Summary:**

This work proposes "Learning from Randomness" (LFR), which tackles the long-standing problem of removing augmentations from SSL. Instead, random projections of the representation are learnt via a bank of predictors, with the intuition that diverse random projections replicate a set of generic downstream tasks.
An objective like Barlow-Twins is used, as well as an iterative training procedure that updates the backbone and the projectors in separate steps. In order to have diverse random projections, the authors propose to sample several of them and select those that are more decorrelated.

The experiments show how the proposed method is a suitable option for time-series, tabular data and medical images datasets. Interestingly, no augmentation is used in all these settings, which I find interesting and novel. An insightful ablation study is also provided.

**Strengths:**

* The problem of removing augmentations in SSL is of great importance in the community.

* The paper is very well written, organized and presented.

* The authors will release code.

**Weaknesses:**

* The experimental section does not include any medium/large scale dataset, which obfuscates the benefit of the method for any SSL task.

* I found a transfer learning experiment lacking in the results.

* More discussion about the pros/cons of using random projections would be valuable.

**Questions:**

I do not have a large amount of questions, since the paper is very well explained and motivated. However, I would greatly appreciate to discuss with the authors about the following:


* The main intuition of this method is to create random downstream tasks that capture different aspects of $z$. Therefore, I would suggest adding a transfer learning experiment, in which an encoder $f_\theta$ is learnt on a source dataset, and such features are transferred to other real downstream tasks. It would be really interesting to see a gain in such setting, wrt. other SOTA methods. I believe this would strongly reinforce the claim of the paper.

* I wonder if random projections are enough to capture the diversity inherent in complex datasets. I think the paper would benefit from a discussion on the pros and cons of using random projections. Note that the datasets shown in the paper are considered small-scale in the community, which limit the understanding on how effective LFR is for _any_ dataset.   Larger, or more diverse, datasets would be extremely valuable. See following point.

* I suggest evaluating LFR vs. computer vision methods in a different scenario than medical images. It is possible that SimCLR would perform better on CIFAR-10 or ImageNet, nevertheless that experiment would show how well LFR is performing against methods tailored for such scenarios. This would also help to understand how well random projections "emulate" the use of well-chosen augmentations. I think a natural images dataset is required for this paper for acceptance.
  * In the Appendix, the authors provide an example of feature interpretability on CIFAR-10. Noting that the pipeline to train on natural images is already in place, I reinforce my observation that an evaluation on natural images should be part of the experimental section.

* It would be great to have some discussion about how LFR could be applied to Transformers, given their wide use. Comparison with a masking approach would not be required, although of great interest.

* A batch size of 256 is strongly detrimental for a contrastive approach such as SimCLR, for example. Small batch sizes harm performance for such methods. For the reader to fully grasp the scenarios where LFR is suitable, I suggest comparing with SimCLR in settings where the latter is known to excel.

* I highly appreciated the breakdown of GPU hours required for this paper.

-----

Overall comment:

The method proposed is sound, and the goal of removing augmentations from SSL is a long-standing one. Learning from random projections seems a sensible way to tackle such problem. The overall architecture and objective is solid, I have no concerns in that sense. The manuscript is written elegantly, with the appropriate language.

However, for the method to be rigorously evaluated, more experiments would be required. Otherwise, there is doubt about whether random projections are valuable for small datasets only, etc.

Notably, I suggest the authors to:
* Perform transfer learning experiments to support the main hypothesis of the work (random projections allow learning different aspects of $z$, thus making $z$ more generally applicable).
* Evaluate in a setting where classic methods perform at their best, to understand the limitations of the proposed approach. I would suggest a natural images setting, using ImageNet ideally, or CIFAR-10/100 if GPU hours are a concern.

* I think it would be interesting for the reader to see how the training behaves with joint training, I suggest adding some of these plots in the Appendix.

I would be happy to increase my score after discussion and manuscript updates.

**Details Of Ethics Concerns:**

None.

---

> ### Author Response · Authors · 2023-11-18
> **Initial Response**
>
> We are pleased that you found the motivation of our methods to be of great importance and our solution to be sensible suggestion. We will try to address each of your points.
>
> **W1: The experimental section does not include any medium/large scale dataset.**
>
> We disagree that there are no large-scale datasets used in our experiments. Please see Table 3 in Appendix C.1 for a summary of datasets and their sizes. We point out that MIMIC-III has a training set with over 2.5M datapoints, and HEPMASS consists of 7M. Meanwhile, our medical image dataset Kvasir consists of images at 80x100 resolution which is a fair bit larger than something like CIFAR10.
>
> In the general comment, we have added experiments on two additional datasets - CIFAR10 for the natural images domain, and a fault diagnosis dataset from (Eldele et al. 2021).
>
> **W1/Q1/S1: Perform transfer learning experiments to support the main hypothesis of the work.**
>
> We appreciate this suggestion and that you think such results would “strongly reinforce the claim of the paper”. We originally did not include transfer learning experiments because they are not straightforward in the tabular and time-series domain. However, since it was a common request, we did find a suitable setting from prior literature and applied LFR to it. The results are discussed in the general comment.
>
> **W3/Q2: More discussion about the pros/cons of using random projections would be valuable.**
>
> We can provide some additional thoughts on the pros and cons of learning from random projections.
>
> While we motivate our work with reference to domains where semantic-preserving augmentations are difficult to craft, there certainly are other domains where they do exist (e.g. natural images). In those cases, LFR does not make use of the expert domain knowledge that is available. This could lead to suboptimal performance, as indicated by our new results on CIFAR-10 in the general comment. One way to improve LFR would be through incorporating this domain knowledge when available.
>
> LFR also requires initializing many networks for random projectors, but then discards many of them when selecting for diversity. This could be seen as wasteful, and the method could be improved if there were a good way to initialize more diverse networks to begin with, but we note that these are small networks, and that the entire process takes a negligible computational time compared to SSRL (3.01s to initialize and select for diversity, vs 849.21s to train).
>
> In terms of pros, we have tried to highlight these throughout the paper. LFR can be used on any domain or modality. LFR has better performance across tasks where semantic-preserving augmentations are not well-developed, including important domains like medical images (Table 2). LFR can be more computationally efficient than methods like SimCLR and SimSiam that use multiple passes through the large encoder and CPU-based augmentations (see general comment).
>
> **Q3/S2:  Evaluate in a setting where classic methods perform at their best. A natural images dataset is required for this paper for acceptance.**
>
> This was a common request amongst reviewers, so we have provided discussion and experimental results on CIFAR10 in the general comment. In summary, LFR (using no augmentations) does not outperform SimCLR (using the full suite of augmentations) on natural image datasets. The same can be said for several domain-agnostic approaches including DIET and DACL.
>
> **Q4: Could LFR be applied to Transformers?**
>
> We believe it is possible to use LFR with transformers, but have not been able to implement this idea in the discussion period. Hopefully the community is able to explore these ideas upon publication. As a brief outline, one could modify contrastive approaches like SimCSE [A] to use LFR by replacing how SimCSE samples positive pairs. Using a pre-trained transformer to transform text inputs to representation space, random projections can be applied instead of sampling pairs, and then SimCSE’s contrastive loss can be replaced with the batch-wise Barlow Twins loss.
>
> [A] Gao et al., “SimCSE: Simple Contrastive Learning of Sentence Embeddings”, EMNLP 2021
>
> **Q5: A batch size of 256 is strongly detrimental for a contrastive approach. I suggest comparing with SimCLR in settings where the latter is known to excel.**
>
> We agree that contrastive methods have been shown to perform well at very large batch sizes. We see this as a negative of such approaches, as it increases the computational requirements of running such methods. Based on Figure 4, LFR does well at reasonably small batch sizes. However, we take your point, so for the new CIFAR10 experiments in the general comment we did increase the batch sizes to 512.
>
> **S3: It would be interesting to see how joint training behaves.**
>
> We did experiment with joint training, with an ablation on Kvasir shown in Figure 4. This ablation helps to justify why we used an EM-like algorithm rather than the simpler joint training idea.

---

> > ### Comment · Reviewer_Eryk · 2023-11-21
> > **Answer to a great rebuttal**
> >
> > I want to thank the authors for a very detailed and well written rebuttal, I definitely enjoyed reading the general answer as well as the dedicated answers to my review.
> >
> > * The authors answer about medium/large scale datasets is accurate, I apologize for the mistake.
> > * The transfer learning experiment is insightful and shows the benefit of LFR. I think this experiment is important for the paper and strengthens it.
> > * I appreciate the experiement on CIFAR-10, where the authors show that cherry-picked augmentations with domain knowledge are performing better. In my opinion, this experiment complements the claims and enriches the paper from a scientific perspective. The fact that SimCLR with dedicated augmentations would perform better was expected, being some sort of "upper bound" for LFR (if we had unlimited capacity to search for the optimal random projectors). I am grateful that this aspect of the method will make it to the main paper.
> > * The remaining answers are also adequate.
> >
> > Given all the above, as well as the detailed answers to the other reviewers, I am willing to upgrade my score.

---

### Official Review · Reviewer_6gYm · 2023-10-30

**Soundness:** 2 fair
**Presentation:** 2 fair
**Contribution:** 2 fair
**Rating:** 6
**Confidence:** 2

**Summary:**

This paper proposed the method of self-supervised representation learning that can be used to learn useful representations from different modalities, e.g. images, text, time series, tabular data. Proposed method is based on recent approach where we have architectures with an additional projector (Guillotine regularization, removed for a downstream task) tries to predict multiple random projectors. The authors provide some analysis of hyper-parameter sensitivity, different initializations of random projectors, etc. Comparison with other methods on different modalities are proposed (time series, tabular, image).

**Strengths:**

S1. The method is simple and very generic. Removes the prior of knowing what data aug. we should use for many current SSL contrastive-based methods.
S2. Paper easy to follow and well-written.

**Weaknesses:**

W1. The main weakness for me is inability to compare with the existing method on well-known datasets in computer vision tasks. We have 3 x time series, 3 x tabular, and 1 x image - where Kvasir is not commonly used dataset. Not sure if the results are not picking the datasets that show good results. Why not presents the results on ImageNet, and if computationally not possible - cifar100.

W2. There's no evaluation on the different downstream task for the learned representation, e.g. feature extractor trained with LFR on Kvasir and evaluated on the other dataset similar and disimilar one. We do not know how the learned representation generalize, and what if we only memorize patterns that then can be then useful for the lin. evaluation.

W3. We don't know the final computational overhead of the initialization and comparison to any SSL method. In the appendix we have total time spent for a particular dataset (e.g. Kvasir V100 1095 GPU hours for all experiments). Would be better to know the comparison between SimCLR/SimSiam vs LFR.

W4. 2 time series datasets (HAR, Epilepsy) and tabular Income&HEPMASS have already good accuracy on the randomized init. What is intresting, some methods are below that (HAR - Autoencoder).

W5. Lack of more theoretical explanation why it should work? What random projectors can be used? etc.

**Questions:**

Q1. How the method perform without using heavy SSL (SimCLR) data augmentations?

---

> ### Author Response · Authors · 2023-11-18
> **Initial Response (1/2)**
>
> We are glad that you appreciated the simplicity and generality of our approach, thank you for your feedback. We will go through each of your points.
>
> **W1: The main weakness for me is inability to compare with the existing method on well-known datasets in computer vision tasks. We have 3 x time series, 3 x tabular, and 1 x image - where Kvasir is not commonly used dataset.**
>
> We must disagree on one aspect - Kvasir is in fact widely used for benchmarking applications of ML to healthcare with over 1000 combined citations [Pogorelov et al. 2017] [A]. We used it as an example of the differences between domains (within the image modality), and how natural image augmentations are not necessarily semantic-preserving for medical images. This is a core motivation of our paper - moving towards domain-agnostic SSRL methods - and was not picked based on results.
>
> As to your interest in seeing results on natural image data as well, and based on multiple requests from reviewers, we have rerun our experimental pipeline on CIFAR10. The results and discussion are in the general comment. In summary, LFR (using no augmentations) does not outperform SimCLR (using the full suite of augmentations) on natural image datasets. The same can be said for several domain-agnostic approaches including DIET and DACL.
>
> **W2: There's no evaluation on the different downstream task for the learned representation. We do not know how the learned representation generalize, and what if we only memorize patterns that then can be then useful for the lin. Evaluation.**
>
> Again, many reviewers had the same thoughts, so we have addressed your idea about transfer learning in our general comment. We originally did not include transfer learning experiments because they are not straightforward in the tabular and time-series domain. However, we did find a suitable setting from prior literature and applied LFR to it.
>
> **W3: We don't know the final computational overhead of the initialization and comparison to any SSL method. In the appendix we have total time spent for a particular dataset (e.g. Kvasir V100 1095 GPU hours for all experiments). Would be better to know the comparison between SimCLR/SimSiam vs LFR.**
>
> The initialization and selection of random projectors is a minor part of overall training time with LFR. As a concrete example, on the Income dataset training LFR for 100 epochs took 849.21s on our machine, while the DPP selection took 3.01s.
>
> In terms of model training, SimCLR took around 9 hours on the Kvasir dataset, while LFR completed in just 2 hours on the same machine. Please see the general comment for a discussion of this result and our analysis of computational costs.
>
> **W4:  2 time series datasets (HAR, Epilepsy) and tabular Income&HEPMASS have already good accuracy on the randomized init. What is intresting, some methods are below that (HAR - Autoencoder).**
>
> The randomly initialized encoder baseline is intended to benchmark the improvement given by the learned representations. If the downstream task is easy, or the input features are already in a useful representation, then linear evaluation could perform well on any representation, even a randomly initialized one. Methods that do not outperform Random Init on linear evaluation are not providing more useful learned representations. In Table 1 there are various instances where an SSL method underperforms this baseline, possibly because the augmentations it uses are not well suited for the domain. However, LFR, being domain-agnostic, is able to learn more useful representations in every setting.

---

> > ### Author Response · Authors · 2023-11-18
> > **Initial Response (2/2)**
> >
> > **W5: Lack of more theoretical explanation why it should work? What random projectors can be used? Etc.**
> >
> > We found it sensible to use random projectors that share a similar architecture to the representation model. For each modality, researchers have already determined classes of network architectures that have good inductive biases for extracting useful features. This information dictates the choice of representation model, and we reuse that information in choosing the projector architecture. In principle one could use any projector, but in preliminary experiments we found that matching inductive biases was beneficial. While we do not have a complete theoretical explanation for the success of learning from randomness, we can offer you some indication of an interesting direction. We anticipate that the research community will be interested in exploring theoretical explanations after publication.
> >
> > Random orthogonal projections on high dimensional spaces have been studied in mathematics and found to have interesting properties. One result, known as the Johnson-Lindenstrauss lemma [B], says that a set of points on a high-dimensional Euclidean space can be embedded into a lower-dimensional space such that distances are preserved. It has been observed that distances are still well-preserved when the embedding is a random orthogonal projection. This rigorous mathematical setting loosely mimics the practical setting of our paper - where high-dimensional data is projected into a low-dimensional space via a randomly initialized network. While the network does not produce an orthogonal embedding, it may still approximately preserve the distance between datapoints. This information can then be leveraged by the representation learner.
> >
> > We do not claim this to be a sufficient theoretical explanation, but we hope to investigate it in future work and look forward to other explanations that the community can propose.
> >
> > **Q1:  How the method perform without using heavy SSL (SimCLR) data augmentations?**
> >
> > We believe you are asking how LFR would perform if it was used *with* heavy data augmentations, since our method does not use any augmentations to begin with. It could be possible to perform data augmentations before sending a datapoint through our encoder and random projectors. Such augmentations could potentially increase the diversity of data seen by the model, but it is important to note that our method is not contrastive in the same way as, for example SimCLR. In SimCLR the augmentations produce two views of a datapoint which are deemed to be semantically the same, and the representations of the views are pushed together. Augmenting data in LFR would not have the same meaning. In effect, the random projectors of LFR are producing multiple views, and the representation model must encode enough information for the predictor heads to match the views. Whether data is augmented or not before being projected does not fundamentally change this process.
> >
> > Since we designed and tested our method for domains where semantic-preserving augmentations are not known or easy to create, we did not conduct a test that uses heavy data augmentations.
> >
> > [A] Jha et al., “Kvasir-SEG: A Segmented Polyp Dataset”, International Conference on Multimedia Modeling 2020.
> >
> > [B] Freksen, “An Introduction to Johnson-Lindenstrauss Transforms” 2021

---

### Official Review · Reviewer_gtio · 2023-10-30

**Soundness:** 3 good
**Presentation:** 4 excellent
**Contribution:** 3 good
**Rating:** 8
**Confidence:** 4

**Summary:**

The authors of this paper propose a technique called Learning From Randomness (LFR), which allows the application of self supervised techniques in arbitrary data domains. The proposed method works by projecting the data into random representations, and then training a model to predict these random representations. The authors show that the resulting model can learn useful representations, even without domain knowledge for the datasets examined.

**Strengths:**

- The paper is very clear and easy to understand. I did not have any issues in grasping the points the authors are trying to make.

- The method proposed is novel, as far as I am aware. It is also very interesting, as learning from random data is not a very explored area of research. As the authors note, the proposed technique enables self supervised learning without the need for domain knowledge, in order to create good augmentations for the data. There is also a clear benefit from using LFR in the datasets examined, without having to rely on complicated techniques.

- The authors perform ablations on the random projectors used, as well as the required diversity of the random representations and the training procedure for the model. I find it interesting that the authors perform an EM-based approach in learning the model, instead of simple optimizing all of its parts all at once. Similarly, I find equally interesting the preprocessing step that selects the best random projectors to predict during training.

Overall, I find the proposed method insightful, with clear benefits over previous work in datasets that are not as explored as the usual ones (e.g. CIFAR-10/100, ImageNet).

**Weaknesses:**

- One of the issues I have with this paper is that despite the use of several datasets used to evaluate LFR, the commonly used ones such as CIFAR-10/100 are not among them. While I understand that LFR does not aim to improve performance on these datasets (since natural image augmentations already perform very well) it would be interesting to examine those to compare as well. It would be useful to know if LFR is better/worse than optimized augmentations, such as those used in SimCLR.

- I think the paper could also be improved via further experiments on the following two subjects:

  - I think it would be interesting to see some ablations on the distance metric used for training. Right now, the authors use Barlow Twins as the metric, but it would be interesting to perform ablations with e.g. MSE or Contrastive losses for this (although I must note that the authors do make an argument for this design decision in the paper).

  - I think it would also be interesting to see the transferability of the trained models across different datasets. I would be interested in knowing if LFR leads to learning representations that are good for the particular dataset, or good for the chosen modality in general.

**Questions:**

I would be grateful if the authors could comment a bit on the choice of the number of projectors $K = 6$ and batch size $B = 256$. Intuitively, both of these values seem a bit small when trying to find diverse random projectors. The authors have tried going up to $K = 8$ and batch size $B = 512$, but I would like to know if they have tried higher values for these two hyperparameters (especially for the number of projectors $K$).

---

> ### Author Response · Authors · 2023-11-18
> **Initial Response**
>
> We are very pleased to see that you enjoyed our paper, and will address the few comments you had.
>
> **W1: While LFR does not aim to improve performance on datasets like CIFAR10, it would be useful to know if LFR is better/worse than optimized augmentations, such as those used in SimCLR.**
>
> This was a common sentiment amongst reviewers, so we have provided discussion and experimental results on CIFAR10 in the general comment. In summary, LFR (using no augmentations) does not outperform SimCLR (using the full suite of augmentations) on natural image datasets. The same can be said for several domain-agnostic approaches including DIET and DACL.
>
> **W2.1: It would be interesting to perform ablations with e.g. MSE or Contrastive losses.**
>
> Since multiple reviewers brought up this point, we have addressed it in the general rebuttal by running experiments using LFR with InfoNCE instead of Barlow Twins. In addition to the arguments for our design decision in the paper, our batch-wise Barlow Twins loss did give a slight improvement over InfoNCE for downstream performance.
>
> **W2.2: It would also be interesting to see the transferability of the trained models across different datasets.**
>
> We originally did not include transfer learning experiments because they are not straightforward in the tabular and time-series domain. However, since it was a common request, we did find a suitable setting from prior literature and applied LFR to it. The results are discussed in the general comment.
>
> **Q1: I would be grateful if the authors could comment a bit on the choice of the number of projectors K=6 and batch size B=256. I would like to know if they have tried higher values for these two hyperparameters (especially for the number of projectors K).**
>
> As you noted, we performed ablations on Kvasir for both number of projectors and batch size in Figure 4 of Section 4.6. The largest values we used were $K=8$, and $B=512$. Already at these levels there was a decrease in final performance, so we have not extended it to larger values of $K$. Generally, our findings are that the projector selection approach from Section 3.3 is adequate for ensuring enough diversity in a small number of projectors that $K$ does not need to be large (which would add some cost to training, specifically from the $K$ predictor heads that try to match the $K$ projectors).

---

> > ### Comment · Reviewer_gtio · 2023-11-20
> > **Response to rebuttal**
> >
> > I'd like to thank the authors for their extensive responses to both mine and the other Reviewers' comments. These provide further insight into how LFR can be used in a variety of domains. Even though in the case of CIFAR-10 the performance is significantly lower than SimCLR/SimSiam, I think that is to be expected, given that image-based augmentations have been very extensively studied.
> >
> > As such, I am keeping my suggestion to accept the paper for now, since I believe the goal of LFR is very interesting and novel.

---

### Official Review · Reviewer_ze2f · 2023-10-31

**Soundness:** 2 fair
**Presentation:** 3 good
**Contribution:** 2 fair
**Rating:** 5
**Confidence:** 4

**Summary:**

This paper considers the problem of domain-agnostic self-supervised representation learning. The proposed method introduces multiple random projectors and corresponding predictors, and optimizes the batch-wise barlow twins loss, which constructs the Gram matrix instead of the empirical correlation matrix. To encourage the diversity, many projectors are initialized and then only 10% are subsampled for use. Experimental results on datasets from various domains show the effectiveness of the proposed method.

**Strengths:**

- Domain-agnostic representation learning is a timely topic.

- The proposed idea is simple and the proposed method improves the performance in most cases.

**Weaknesses:**

- Discussion/comparison with other domain-agnostic methods seem to be not enough. For example, [Lee et al.] proposed a domain-agnostic augmentation strategy applied to image, speech, and tabular datasets, and [Wu et al.] proposed randomized quantization and experimented on image, point cloud, audio domains and the DABS benchmark. I suggest including discussion and experimental comparison with them.

- It is good to see that experiments include various domains including time series, tabular, and image, but they seem to be relatively small and not commonly used for benchmarking machine learning models. For example, Kvasir is a medical image dataset, which is different from the widely used "natural" image datasets; it should be categorized differently from natural image datasets. Authors may want to refer to [Lee et al.] and [Wu et al.] to find commonly used datasets to provide the general applicability to various domains and scalability of the proposed method.

- While the authors claim that the optimization strategy for the proposed method is EM, but it is not clear how the proposed alternating optimization is related with EM by looking at the formulation. I think the transition from Eq. (2) to Eq. (3--4) requires more explanation supported with some math.

- The claim around batch-wise barlow twins that MSE is preferred over cross-entropy/contrastive/triplet losses is not justified. Isn't the batch-wise barlow twins loss just a kind of contrastive loss, in that it contrasts all samples within the batch? Note that the original contrastive loss (not the InfoNCE variation) also computes the MSE loss. An ablation study with different type of losses might also be helpful.

- The criterion for diversity encouragement requires more intuition. It is hard to imagine what is going on when optimizing the proposed learning objective. Also, what is the computational cost for the NP-hard objective function?

- The comparison might not be fair as the proposed method requires more computational cost to encode input with multiple random projectors and predictors compared to other baselines. The computational cost should be matched for a fair comparison and reported.

[Lee et al.] i-Mix: A Domain-Agnostic Strategy for Contrastive Representation Learning. In ICLR, 2021.

[Wu et al.] Randomized Quantization: A Generic Augmentation for Data Agnostic Self-supervised Learning. In CVPR, 2023.

**Questions:**

Please address concerns in Weaknesses.

> **post rebuttal**

After discussion with authors, I feel that the experimental results are not sufficient to support the claim that they cover "a wide range of representation learning tasks that span diverse modalities and real-world applications." Initially their experiments covered time series, medical image, and tabular domains, and the additional results in the natural image domain show that their method is not effective for natural images, compared to other baselines. **Authors are encouraged to explicitly limit the scope to the domains they experimented in the title/abstract/intro.**

Also, I am not sure if the comparison is fair (e.g., if they tuned hyperparameters for baselines properly), so experimental results are generally not convincing to me.

Though I feel more confident on my rating, given that authors addressed all concerns from the other reviewers well and the proposed method is still interesting to me, I do not want to put too much weight to my rating.

---

> ### Author Response · Authors · 2023-11-18
> **Initial Response**
>
> Thank you for your constructive review, we are happy to address each of your points.
>
> **W1 Discussion/comparison with other domain-agnostic methods seem to be not enough:**
>
> While we used three approaches to domain-agnostic SSRL (along with five strong domain-specific baselines), the methods you point out are also useful and we will cite them with discussion. Random Quantization (RQ) [Wu et al. 2023] has received attention with its recent publication at ICCV 2023 in October, which post-dates our submission to ICLR. Nevertheless, this is a helpful suggestion, so we re-ran SimCLR and SimSiam using RQ augmentations on three datasets, one from each modality. However, we found i-Mix [Lee et al.] to be very similar to the more recent DIET [Balestriero 2023], and given that it was shown to be outperformed by RQ [Wu et al. 2023], we believe our addition of a RQ baseline sufficiently covers these alternatives.
>
> The results for SimCLR, SimSiam, and LFR are reproduced from Table 1, with new results added for RQ augmentations. Note that for the Income baseline we restated the SCARF result, since it is similar to SimCLR with random feature corruptions as the augmentation. RQ appears beneficial on HAR (Time Series), but is not necessarily helpful on Income (Tabular) and Kvasir (Image). We note that when Wu et al. applied RQ to image data, they combined it with more traditional image augmentations, as using RQ alone did not produce great performance (Tables 3, 4 in [Wu et al. 2023]). Ultimately, LFR still outperformed this baseline.
>
> |Method/Dataset| HAR | Income | Kvasir|
> |---|---|---|---|
> |SimSiam| 65.1 | 79.2 | 72.6|
> |SimSiam-RQ| 78.9| 76.4 | 73.1|
> |SimCLR (*SCARF)|87.8| *84.2| 72.1|
> |SimCLR-RQ| 91.5 | 78.6 | 68.6|
> |LFR|93.1 | 85.2 | 74.9|
>
> **W2 Datasets are relatively small and not commonly used for benchmarking machine learning models.**
>
> We respectfully disagree on this point. Table 3 in Appendix C.1 summarizes all the datasets we used, which includes MIMIC-III at over 2.5M training examples, and HEPMASS at 7M. Compared with common datasets like CIFAR-10, these datasets are very large scale. The smaller datasets, HAR and Epilepsy, were chosen because they have previously been used for benchmarking time-series SSRL methods (e.g. Eledele et al. 2021), while Income and Theorem have been used for tabular benchmarking (e.g. Hajiramenzanali et al. 2022). In fact these datasets are commonly used: the paper introducing MIMIC-III has over 6000 citations [A], Kvasir has over 1000 combined [Pogorelov et al. 2017] [B], HAR over 2000 [Anguita et al. 2013], Epilepsy over 3000 [Andrzejak 2001], and Income over 2000 [Kohavi 1996]. As a side note, we did explicitly call out that Kvasir was a medical image dataset (not natural image) in Table 2 and Section 4.1.
>
> You also mentioned two domain-agnostic works with more suggestions for other datasets to consider. [Lee et al.] uses image, speech, and tabular datasets, while [Wu et al.] experiment on images, point clouds, and audio. As can be seen, there is a lack of consistency in which modalities are tested for domain-agnostic work, and unfortunately we are not able to introduce several new modalities in the discussion period. However, to partially accommodate your requests, we have repeated our experimental setup on CIFAR-10 as a representative natural image dataset with the results shown in the general comment. The DABS benchmark does appear useful, and we will investigate it for the future.
>
> [A] Johnson et al., “MIMIC-III, a freely accessible critical care database”, Nature Scientific Data 2016.
> [B] Jha et al., “Kvasir-SEG: A Segmented Polyp Dataset”, International Conference on Multimedia Modeling 2020.

---

> > ### Author Response · Authors · 2023-11-18
> > **Part 2**
> >
> > **W3 The transition from Eq. (2) to Eq. (3--4) requires more explanation.**
> >
> > Consider the MLE objective in Equation 2. As variable $\mathbf{z}_i$'s distribution could be intractable, we introduce a manageable distribution $q(\mathbf{z}_i)$ such that
> >
> > $\sum_i\sum_k \log \int_{\mathbf{z}_i}  p\left(\mathbf{y}_i^{(k)}\Bigm| \mathbf{z}_i\right) p\left(\mathbf{z}_i\mid \mathbf{x}_i\right)   d\mathbf{z}_i = $
> >
> > $ \sum_i\sum_k \log \int_{\mathbf{z}_i} q(\mathbf{z}_i)\frac{ p\left(\mathbf{y}_i^{(k)}\Bigm| \mathbf{z}_i\right) p\left(\mathbf{z}_i\mid \mathbf{x}_i\right) }{q(\mathbf{z}_i)}d\mathbf{z}_i $
> >
> > Using Jensen's inequality, the lower bound of the objective is therefore
> > $\sum_i\sum_k \log \int_{\mathbf{z}_i} q(\mathbf{z}_i)\frac{ p\left(\mathbf{y}_i^{(k)}\Bigm| \mathbf{z}_i\right) p\left(\mathbf{z}_i\mid \mathbf{x}_i\right) }{q(\mathbf{z}_i)}d\mathbf{z}_i \geq$
> >
> > $\sum_i\sum_k \int_{\mathbf{z}_i} q(\mathbf{z}_i)\log\frac{ p\left(\mathbf{y}_i^{(k)}\Bigm| \mathbf{z}_i\right) p\left(\mathbf{z}_i\mid \mathbf{x}_i\right) }{q(\mathbf{z}_i)}d\mathbf{z}_i.$
> >
> > Hence, to maximize the variational lower-bound of the above equation with respect to the proposed distribution $q(\mathbf{z}_i)$, we note its optimal solution of $q(\mathbf{z}_i)$ is simply letting the equation hold (as in the classic EM algorithm) such that
> >
> > $\sum_i\sum_k \log \int_{\mathbf{z}_i} q(\mathbf{z}_i)\frac{ p\left(\mathbf{y}_i^{(k)}\Bigm| \mathbf{z}_i\right) p\left(\mathbf{z}_i\mid \mathbf{x}_i\right) }{q(\mathbf{z}_i)}d\mathbf{z}_i =$
> >
> > $\sum_i\sum_k \int_{\mathbf{z}_i} q(\mathbf{z}_i)\log\frac{ p\left(\mathbf{y}_i^{(k)}\Bigm| \mathbf{z}_i\right) p\left(\mathbf{z}_i\mid \mathbf{x}_i\right) }{q(\mathbf{z}_i)}d\mathbf{z}_i,$
> >
> > which, in turn, requires the following equation to hold
> > $\frac{ p\left(\mathbf{y}_i^{(k)}\Bigm| \mathbf{z}_i\right) p\left(\mathbf{z}_i\mid \mathbf{x}_i\right) }{q(\mathbf{z}_i)} = C$, where $C$ is an constant. By transforming this equation a bit, the optimal solution of $q(\mathbf{z}_i)$ is
> >
> > $q(\mathbf{z}_i) = \frac{ p\left(\mathbf{y}_i^{(k)}\Bigm| \mathbf{z}_i\right)}{ p\left(\mathbf{y}_i^{(k)}\Bigm| \mathbf{x}_i\right)} p\left(\mathbf{z}_i\mid \mathbf{x}_i\right).$
> >
> > As both $p\left(\mathbf{y}_i^{(k)}\Bigm| \mathbf{z}_i\right)$ and $p\left(\mathbf{y}_i^{(k)}\Bigm| \mathbf{x}_i\right)$
> >
> > are delta distributions with probability 1 conditioned on deterministic functions modelled by $g^{(k)}$ and $h^{(k)}_\phi$ respectively, the optimal solution of $q(\mathbf{z}_i)$ is simply $p(\mathbf{z}_i|\mathbf{x}_i)$ given the conditions are satisfied. In other words, we need $p(\mathbf{z}_i|\mathbf{x}_i)$ that can let $\mathbf{y}_i^{(k)}$ become
> >
> > $h^{(k)}_{\phi}(\mathbf{z}_i) $ for all $k$. Thus, the optimization is essentially an EM algorithm where we optimize $\theta$ and $\phi$ alternatively to gradually increase the conditional likelihood of Eq 2.
> >
> > **W4  The claim around batch-wise barlow twins that MSE is preferred over cross-entropy/contrastive/triplet losses is not justified. An ablation study with different type of losses might also be helpful.**
> >
> > Fortunately, the core method of LFR, using random projections of data as a pre-training task, can be used with many different losses. We advocated for the batch-wise Barlow Twins loss because of its ability to learn disentangled representations [Zbontar et al. 2021]. We agree that Barlow Twins could be viewed as a contrastive loss, and did not claim otherwise in the paper, and also agree that a comparison could provide more insight. To this end, we used the InfoNCE loss for two datasets to show that batch-wise Barlow Twins leads to empirical improvements - our new results are discussed in the general rebuttal.
> >
> > **W5 The criterion for diversity encouragement requires more intuition. What is the computational cost for the NP-hard objective function?”**
> >
> > Although the problem is NP-hard in theory, we only need to solve an instance of it with finite size. Compared to the cost of training an encoder, the diversity selection procedure using the Fast Determinantal Point Process (Chen et al. 2018) is an almost negligible contribution to the total training time. As a concrete example, on the Income dataset training LFR for 100 epochs took 849.21s on our machine, while the DPP selection took 3.01s.
> >
> > **W6 The proposed method requires more computational cost to encode input with multiple random projectors and predictors compared to other baselines.**
> >
> > Upon examining the details of each training method, it becomes clear that LFR actually requires *less* computation than alternatives like SimCLR and SimSiam. Please see our general comment for a full discussion on this point.
> >
> > Many SSRL methods incorporate small predictor heads, so LFR is not unique in this regard. Meanwhile, the random projectors that LFR introduces are not a source of ongoing computational cost; since they are fixed at the start of training we perform a single forward pass over the entire dataset and reuse these results throughout training.

---

> > > ### Comment · Reviewer_ze2f · 2023-11-22
> > >
> > > Thanks for addressing my questions! Below I provide more comments.
> > >
> > > > W1 Discussion/comparison with other domain-agnostic methods seem to be not enough:
> > >
> > > > W2 Datasets are relatively small and not commonly used for benchmarking machine learning models.
> > >
> > > My major concern is still on this, and the experimental results are not sufficient to claim your method is effective across "a wide range of representation learning tasks that span diverse modalities and real-world applications." I agree that papers on domain-agnostic method such as DACL, i-Mix, and RQ do not have a consensus on the choice of domains, but at least the natural image domain is common, and you can take at least one of the prior works' setting rather than using completely different domains than others to make experimental results more convincing.
> > >
> > > Or, as I suggested in another thread, if you are not willing to incorporate them, then I think it is better to explicitly limit your focus/contribution (in the title/abstract/intro) to medical image and/or tabular domains, and explain why the proposed method is specifically useful for such domains (if possible).
> > >
> > >
> > > > W3 The transition from Eq. (2) to Eq. (3--4) requires more explanation.
> > >
> > > Thanks for the details, hopefully you can incorporate this at least somewhere in the appendix.
> > >
> > > > W5 The criterion for diversity encouragement requires more intuition. What is the computational cost for the NP-hard objective function?”
> > >
> > > Thank you for your response and it is good to know that the NP-hard problem is lightweight to solve in practice. However, still **it is hard to imagine what is going on when optimizing the proposed learning objective** for me.
> > >
> > > > W6 The proposed method requires more computational cost to encode input with multiple random projectors and predictors compared to other baselines.
> > >
> > > Based on the discussion in another thread, unfortunately I feel that prior works are experimented in a suboptimal way (e.g., using suboptimal hyperparameter choices), that might result in an unfair comparison.

---

> > > > ### Author Response · Authors · 2023-11-22
> > > > **Discussion with Reviewer ze2f**
> > > >
> > > > Thank you for clarifying your points and updating your position.
> > > >
> > > > We maintain that our claim is accurate and not overreaching, as we have shown that the method is effective on various domains within three modalities. We did not claim that the method can match augmentation-heavy approaches on the natural images domain, and have provided clear evidence to the contrary. We did use datasets from multiple prior works in SSRL, just not from the three specific papers you mention, which unfortunately all use different baselines with little to no overlap
> > > >
> > > > We are glad that you found the additional steps of derivation helpful, and we can certainly include them in the next update to the paper.
> > > >
> > > > The diversity encouragement method uses a detrimental point process on the projected representations of a batch of data. It selects random projectors that produce maximally diverse representations as given by the determinant of a matrix constructed from those representations. For an in-depth discussion on DPPs and diversity encouragement, please see [A].
> > > >
> > > > We have answered your question on why using multiple random projectors does not incur heightened computational costs compared to baselines - random projectors are not trained or backpropagated through, so we can pre-compute their representations once and reuse them throughout training. We used the same key hyperparameters across methods, including architectures and batch size. Based on prior work, we agree that performance could be improved on datasets like CIFAR-10 by greatly increasing model size, batch size, and training epochs, but we do not have the resources at this time to match such settings. While we agree that a full empirical review of all domain-agnostic SSRL methods across a wide range of baselines, each with a range of architectures and extensively tuned hyperparameters, would be a beneficial resource to the community, such a sweeping review of the topic is beyond the scope of our work and beyond the resources we have available.
> > > >
> > > > [A] Kulesza and Taskar, “Determinantal point processes for machine learning”, Foundations and Trends in Machine Learning 2012

---

### Author Response · Authors · 2023-11-18
**General Comment (Part 1)**

We thank all of the reviewers for their time and effort. We were extremely pleased that reviewers praised the direction of our work: **ze2f** found the topic to be “timely”, **Eryk** deemed the problem we address to be “long standing” and of “great importance”, while **gtio** noted our method’s “clear benefits over prior work”, as well as its novelty. In terms of our results, **sQMY** commented on their “extensiveness”, and together with **ze2f** remarked on the good performance of our method. Finally, several reviewers appreciated the clarity of our paper, including **6gYm**, **gtio**, and **Eryk** calling it “very well-written”.

We noted some commonalities in the questions asked, so we address these topics together.

**Transfer Learning:** Several reviewers asked about transfer learning experiments (**gtio**, **6gYm**, and **Eryk**). Our experimental focus was on the time series and tabular modalities since it is challenging to craft useful augmentations on these data types. Coincidentally, these modalities are usually not amenable to transfer learning. Unlike image datasets which can be cropped into a uniform format, for tabular datasets the number of columns are not consistent across datasets, nor are their meanings. Therefore, an encoder cannot simply be applied to other datasets. A recent survey paper on tabular deep learning notes that “there are no generally accepted ways to do transfer learning for tabular data” [A]. This is why we did not originally include such an experiment.

Nevertheless, since this topic was so widely requested amongst reviewers we build on the experimental setup of (Eldele et al. 2021), one of the very few prior works that performs transfer learning on tabular data. This paper used a set of four real-world fault diagnosis datasets (A, B, C, D) from the same domain and with the same format, but with different characteristics. Hence, we train a representation model on one of the four, and transfer it to the others.

In the table below, the first two rows are reproduced from (Eldele et al. 2021) where the TS-TCC method with a transformer architecture was trained on one task, and then finetuned with 1% of the data on the transfer task. This had been compared to a supervised setting. In our experiments we used the same finetuning scheme, and reused the 3-block CNN architecture that we had applied to HAR and Epilepsy, with similar training hyperparameters (see Table 5 in Appendix C.3).

From our new results, we see that LFR is able to learn features on the source task that transfer well to related tasks, outperforming the supervised approach (with the same model architecture) on 10/12 dataset pairs, TS-TCC on 8/12 pairs, and is significantly better than all other approaches in terms of average performance. Hence, we see that LFR is able to transfer representations in a tabular setting, at least when there is some similarity in the domain of the datasets used.

|Method / Transfer Task| A->B|A->C|A->D|B->A|B->C|B->D|C->A|C->B|C->D|D->A|D->B|D->C|AVG|
|-|-|-|-|-|-|-|-|-|-|-|-|-|-|
| Supervised (Transf.) |34.28|44.94|34.57|52.93|63.67|99.82|52.93|84.02|83.54|53.15|99.56|62.43|63.83|
|TS-TCC (Transf.)|43.15|51.50|42.74|47.89|70.38|99.30|38.89|98.31|99.38|51.91|99.96|70.31|67.83|
|Supervised (CNN)|42.96|46.33|46.99|43.55|70.53|94.94|48.50|78.34|74.34|52.71|99.05|70.20|64.04|
| Rand Init (CNN)|79.55|68.80|79.95|78.96| 58.39 | 81.34 | 70.60  | 84.93 | 80.02 | 80.10  | 80.13 | 57.77 | 75.05 |
| LFR (CNN) | 90.51 | 81.63 | 93.40| 75.99 | 75.48 | 88.64 | 69.32 | 80.54 | 88.34 | 78.92 | 87.90| 75.26 | 82.16 |

[A] Borisov et al., "Deep Neural Networks and Tabular Data: A Survey," IEEE Transactions on Neural Networks and Learning Systems.

**Ablation on Loss:** Reviewers **ze2f** and **gtio** both asked about the choice of the batch-wise Barlow Twins loss, and if other SSL losses could similarly be used with LFR. Indeed, losses like InfoNCE can directly be used with our general method of learning from randomness, and below we show a comparison of Barlow Twins (as in the paper), against a new result on InfoNCE [B] for Kvasir and CIFAR-10. Both loss functions yield similar outcomes, with a slight advantage observed for the Barlow Twins loss. This shows that the LFR method is generalizable, and we look forward to how the community extends it to new losses, but also that our novel Batch-wise Barlow Twins loss is a non-trivial improvement over off-the-shelf choices. The proposed Batch-wise Barlow Twins loss offers increased computational efficiency because it relies on straightforward operations such as summation, subtraction, and multiplication. In contrast, InfoNCE involves more resource-intensive operations like exponentiation, logarithm, and divisions.

| | Kvasir | CIFAR-10|
|-|-|-|
| Batch-wise Barlow Twins | $74.9\pm0.6$ | $64.3\pm0.1$ |
| InfoNCE | $72.6\pm0.4$ | $64.1\pm0.1$ |

[B] van den Oord et al. “Representation Learning with Contrastive Predictive Coding” arXiv preprint 1807.03748

---

> ### Author Response · Authors · 2023-11-18
> **General Comment (Part 2)**
>
> **Natural Images**: Throughout the literature on SSRL, natural images are a common testbed, so it is unsurprising that several reviewers were curious about the performance of LFR on a dataset like CIFAR-10 (**ze2f**, **gtio**, **6gYm**, **Eryk**). Since our work is about domain-agnostic representation learning, mainly in cases where it is not clear how to usefully augment the data, we focused on tabular, time-series, and medical imaging applications. Natural images, being a domain where semantic-preserving augmentations are plentiful, were not the intended target of our work. Still, it is understandable why the reviewers are interested in this topic, so we repeated our benchmarking on the CIFAR-10 dataset.
>
> For CIFAR-10 we used the same experimental settings as for Kvasir in our paper (see Appendix C). As a brief recap, each method used ResNet-18 encoders, and 4-layer MLP predictors. For LFR, the projectors were randomly initialized 4-layer CNNs. Based on Reviewer **Eryk**’s comment that baseline methods like SimCLR benefit from large batch sizes, we increased this to 512 from 256 as used for Kvasir. The table below shows that, as one might expect, SimSiam and SimCLR achieve the best performance out of our baselines thanks to their extensively engineered pipeline of image augmentations. Two domain-agnostic approaches that do not use image augmentations, DIET and our LFR, fall into a second tier, outpacing the third tier of DACL, Autoencoder, and Random Init. This result is in stark contrast to what we found on Kvasir, where all methods ended up in a tight range, with LFR at the top. It seems clear that the heavy augmentations of SimSiam and SimCLR really are specialized for natural images, and are not suitable for other domains within the image modality.
>
> | Method      | Kvasir (From Paper) | CIFAR-10 (New) |
> |-------------|---------------------|----------------|
> | Supervised  | $83.2\pm0.2$        | $94.4\pm 0.2$  |
> | Random Init | $28.9\pm5.7$        | $34.7\pm0.3$   |
> | Autoencoder | $72.4\pm0.6$        | $37.4\pm0.3$   |
> | DIET        | $71.3\pm0.9$        | $69.3\pm1.2$   |
> | SimSiam     | $72.6\pm1.4$        | $89.2\pm0.1$   |
> | SimCLR      | $72.1\pm0.3$        | $86.7\pm0.5$   |
> | DACL        | $72.0\pm0.1$        | $42.4\pm 2.7$  |
> | LFR (Ours)  | $74.9\pm0.6$        | $64.3\pm0.1$   |
>
> **Computational Costs:** Reviewers **ze2f** and **6gYm** asked about comparisons of computational costs between the proposed LFR, and baseline methods, especially with regards to the initial cost to select diverse random projectors using the DPP algorithm which is NP-hard. In brief, our experiments across diverse datasets consistently reveal that LFR's training time is often comparable to, or even notably shorter than the contrastive SSRL methods. For example, on Kvasir training SimCLR took around 9 hours, while LFR completed in just 2 hours on the same machine. This assessment of training time considers both CPU and GPU usage.
>
> The efficiency in LFR's training time can be attributed to two primary factors. First, existing SSRL methods rely on extensive augmentations that are performed on CPU and can lead to data loading bottlenecks. LFR has no such issue. CPU-based image augmentations account for a significant amount of SimCLR’s 9 hour training time on Kvasir. For datasets with less computationally intensive augmentations, training times tend to align more closely - on the HAR dataset, LFR concluded training in approximately 3 hours while SCARF required around 3.3 hours.
>
> Second, multiple passes through predictor heads in LFR is not overly burdensome. Unlike contrastive frameworks that use double augmentation and encoder passes, LFR requires fewer expensive operations: one pass through each random projector before training, and one pass through the encoder followed by one pass through each small predictor during training. Given that encoder updates are the most expensive, LFR's training time often proves less than SimCLR.
>
> Compared to the cost of training an encoder, the diversity selection procedure using the Fast Determinantal Point Process (Chen et al. 2018) is an almost negligible contribution to the total training time. To recap, to select $K$ diverse projectors we initialize $N$ small neural networks and for each we run one forward pass on a batch of training data of size $m$. Then, DPP is applied over matrices of size $m^2\times K$ constructed from the network outputs. As a concrete example, on the Income dataset training LFR for 100 epochs took 849.21s on our machine, while the DPP selection took 3.01s.

---

> ### Comment · Reviewer_ze2f · 2023-11-21
> **Some results including other DA methods are far from the original paper, so the claim might be invalid**
>
> Thank you for providing CIFAR-10 results, but some numbers are far lower than those reported in other papers, so the results here are not so convincing to me.
>
> Specifically, below are some comparison of replication here vs. original DACL paper:
>
> SimCLR: 86.7 vs. 93.4
>
> DACL: 42.4 vs. 81.3 (83.8 for DACL+)
>
> Also, this observation is indirectly agreed with [Lee et al.], which reported the performance of a similar mixup-based method without image augmentations as 83.4 on CIFAR-10. This contradicts to your claim that LFR outperforms DACL. At this point, it seems the proposed LFR is not better than other DA methods for some datasets (at least CIFAR-10) under the same condition that no data augmentation is used.
>
> (Edit: I just realized that they used ResNet-50 while you used ResNet-18, but still I don't think that explains the reason why the performance of DACL drops almost 40%p.)
>
> Regarding the claim that "CPU-based image augmentations account for a significant amount of SimCLR’s 9 hour training time on Kvasir (while LFR completed in just 2 hours on the same machine)," it seems you did not properly parallelize data augmentation; data augmentation is generally not a bottleneck of training time for image classification, based on my personal experience on contrastive learning on image datasets. The size of images in Kvasir is only 100x80 as stated in the end of Section 4.1, so I think the claim that SimCLR takes a longer training time than your proposed method due to extensive data augmentation is not so valid.
>
> The reason why your experimental results are not convincing is due to the lack of fair comparisons on the datasets already used in prior works. Hence, I suggest authors to try out those used in prior works like [Lee et al.] or [Wu et al.] including CIFAR-10, and make a fair comparison with them to claim the superiority of the work. (I found that [Wu et al.] is published in ICCV2023 but available since Dec 2022, so it was online for almost an year. I believe this has been exposed for enough time to be recognized if you properly surveyed related works. Or, at least [Lee et al.] published in ICLR2021 is old enough and should be a good reference.)
>
> Also, similar to DACL or [Lee et al.], it is good to try out both (domain-specific) augmentations and the proposed method together and see if it results in further improvements, as there is no reason not to use data augmentations if we know they are effective.

---

> > ### Author Response · Authors · 2023-11-22
> > **Response to Reviewer ze2f**
> >
> > **“Thank you for providing CIFAR-10 results, but some numbers are far lower than those reported in other papers…”**
> >
> > Thank you for engaging on these topics. We emphasize that natural image tasks are not the main point of our work, and have only been included in the discussion phase at the request of reviewers. In our paper we never claimed LFR would outperform on natural image tasks. In fact, we said “if one knows the application domain well with adequate intuition around sensible data augmentations, using contrastive-learning-based SSRL is still likely to outperform random projectors”. Please see our Conclusion section. The new CIFAR-10 results confirm that intuition - LFR is not as good as SimCLR on CIFAR-10, but neither is DACL nor DIET when these methods are used without standard image augmentations.
> >
> > Your concern seems to be the relative performance of domain-agnostic methods on CIFAR-10. As you mentioned, there are a number of differences between the hyperparameter settings we used and what DACL used. This is unfortunately unavoidable. We are trying to make a fair comparison to four prior methods (DACL, DIET, SimSiam, SimCLR). This means, for example, using the same encoders and batch size across tasks, even though individual prior works made different choices. On CIFAR-10, for each method the encoder was a ResNet-18 with batch size 512, trained for 400 epochs with SGD. As one example, when DACL reported a performance of 81.3, they used a ResNet-50(4x width), batch size 4096, 1000 epochs with a LARS optimizer. This is a significant increase in model size and compute, which could explain the discrepancy you pointed out. We are unable to acquire sufficient computing resources and perform such long trainings across multiple methods during the discussion phase.
> >
> > **“...it seems you did not properly parallelize data augmentation… the claim that SimCLR takes a longer training time than your proposed method due to extensive data augmentation is not so valid”**
> >
> > We did not claim that SimCLR takes longer to train because of data augmentations alone. In the “Computational Costs” section of our general comment we outlined two main differences. CPU-based data augmentations are one, while the second has to do with the number of passes through the encoder which is always the largest component of an SSRL training system. SimCLR uses two forward/backward passes per datapoint; LFR uses one pass. Both of these factors are relevant for the overall training time comparison. The comparison we mentioned shows that LFR is not worse in terms of computational costs, which is what some reviewers suspected. For the sake of reproducibility, we have provided our code in the supplementary material.
> >
> > **“The reason why your experimental results are not convincing is due to the lack of fair comparisons on the datasets already used in prior works…”**
> >
> > Many of the datasets we used were chosen because previous SSRL works used them. HAR, Epilepsy, and Fault Detection were used by the TS-TCC paper [Eldele et al., 2021]; Income and Theorem were used by STab [Hajiramezanali et al., 2022]; CIFAR-10 is widely used.
> >
> > Unfortunately, there is no consistency to the datasets used in this field, and we are unable to benchmark on every dataset used in every relevant paper. You are suggesting two relevant works, i-Mix [Lee et al.] and RQ [Wu et al.], but even these two papers only share a single dataset (ImageNet) in common! Given constraints, we made a reasonable decision to select a subset of well-known datasets across three modalities and many domains.
> >
> > As you requested, we did perform new experiments with RQ on three modalities, and presented them in the direct response to your review. Are you simply asking us to also apply RQ to CIFAR-10? In fact, that paper itself did not benchmark on CIFAR-10. We are not sure what more is to be learned, as we have already presented a negative result on CIFAR-10 and are not claiming that LFR is superior in the natural image modality. i-Mix is a relevant work we can mention, but we already have presented a baseline that is built on mixup (DACL), and one that uses data indices (DIET). We have not made any claims that our method is superior to i-Mix.
> >
> > **“... it is good to try out both (domain-specific) augmentations and the proposed method together and see if it results in further improvements…”**
> >
> > DACL and i-Mix are both methods of generating augmentations in a domain-agnostic way, so that those augmentations can be used in a standard contrastive SSRL pipeline. Hence, it makes sense to also add in domain-specific augmentations when available.
> >
> > Our method does not work the same way. We are not creating domain-agnostic augmentations, we are creating domain-agnostic pre-training tasks. Each task is to match the output of a random projector. It is not straightforward to merge an augmentation pipeline into our proposed method.

---

> ### Comment · Reviewer_ze2f · 2023-11-22
>
> Thanks for your quick response! Below I provide more comments, that would hopefully be helpful regardless of the acceptance of this paper.
>
> > We emphasize that natural image tasks are not the main point of our work ..
>
> I think it is inevitable to incorporate the CV (with natural images) and NLP domains if you claim the proposed method works across "a wide range of representation learning tasks that span diverse modalities and real-world applications", as CV and NLP are two major real-world applications that deep learning has been successfully applied.
>
> Rather, if you are not willing to incorporate them, then I think it is better to explicitly limit your focus/contribution (in the title/abstract/intro) to medical image and/or tabular domains, and explain why the proposed method is specifically useful for such domains (if possible).
>
> Regarding the low performance of DACL, generally speaking, if you cannot replicate a prior work on your side, then you can implement your method on top of the prior work's codebase. If the computational cost matters, then you can simply replace the arch with ResNet-18 on the official DACL code, or just go with i-Mix, as it provides ResNet-18 results. It is my surprising that you missed i-Mix, as I can find it within top-5 results by googling one of your keywords "domain-agnostic representation learning"
>
> Regarding your statement **"We are trying to make a fair comparison ..., for example, using the same encoders and batch size across tasks,"** note that using the same hyperparameter does not mean that the comparison is fair. Hyperparameters should be optimized per method via a proper validation process. Rather, you can match some properties/conditions that you believe they are important for a fair comparison, e.g., the wall-clock training/inference time.
>
> > SimCLR uses two forward/backward passes per datapoint; LFR uses one pass. Both of these factors are relevant for the overall training time comparison. The comparison we mentioned shows that LFR is not worse in terms of computational costs, which is what some reviewers suspected.
>
> If you concern two forward/backward passes, then you can remove the mirrored loss in SimCLR (AFAIK this is better than reducing the number of iterations by a half), or just use the momentum encoder like MoCo/BYOL instead. It would be helpful to compare methods in a 2D graph, where the x-axis is the wall-clock training time and the y-axis is the performance.
>
> > Your answer to “The reason why your experimental results are not convincing is due to the lack of fair comparisons on the datasets already used in prior works…”
>
> Again, if you cannot afford with the heavy experimental setting of DACL, then you can try i-Mix with ResNet-18. I have not thought additional experiments on datasets used in i-Mix or RQ can be done within the rebuttal period, so that's one of the main reason why my rating is not positive though I feel the proposed method is interesting.
>
> > It is not straightforward to merge an augmentation pipeline into our proposed method.
>
> Is there a reason why you cannot apply data augmentation in your framework? Data augmentation is generally independent to the model architecture. Or, have you observed that naively applying data augmentation does not improve the performance of the proposed method, like masked autoencoders? I think that's fine, and providing the information about such fact might be useful to audiences.

---

> > ### Author Response · Authors · 2023-11-22
> > **Additional Response to Reviewer ze2f**
> >
> > Thank you for your continued advice on how to improve the paper. We feel that we have made our position clear at this point in the discussion. The claim we made that “the proposed method works across a wide range of representation learning tasks that span diverse modalities and real-world applications" is accurate, not overreaching, and supported by experiments on many domains across three modalities. We have incorporated your advice to test on natural images (CIFAR-10), include other domain-agnostic baselines (experiments on RQ), and perform transfer learning. We can discuss i-Mix as a related work, but we have already benchmarked with recent methods that are very similar (DACL and DIET). Finally, our aim is not to integrate data augmentations with domain-agnostic SSRL, but to build an SSRL method that does not use augmentations at all. This is what we have presented in our paper. Adding domain-specific augmentations back into the framework can be a problem for the research community to tackle if a well-suited problem emerges.

---

### Meta-Review · Area_Chair_83Cg · 2023-12-08

**Metareview:**

This paper introduces a self-supervised representation learning approach by learning high-quality data representations through reconstructing random data projections. This method is applicable across various data modalities and network architectures without relying on augmentations or masking. The paper has been evaluated based on its strengths and weaknesses as identified by the reviewers:

Strengths:

+ Domain-Agnostic Approach: The paper addresses a timely topic of domain-agnostic representation learning, a significant contribution considering the current trends in SSRL.

+ Novel and Simple Methodology: The proposed method is both novel and straightforward, showing improvements in performance across most cases tested.

+ Inclusivity of Various Domains: The experiments include diverse domains like time series, tabular, and image data, showcasing the method's applicability in different contexts.

+ Clarity of Presentation: The paper is commended for its clear and understandable writing, making it accessible to a wide audience.

+ Insightful Ablation Studies: The authors conduct comprehensive ablation studies on random projectors and the training procedure, providing deeper insights into the method's functioning.

+ Theoretical Underpinning: The method's theoretical basis adds depth to the research.

+ General Utility: The simplicity and generic nature of the method are highlighted, removing the need for domain-specific data augmentation knowledge.

Weaknesses:

- Lack of Comparison with Common Benchmarks: The paper fails to include common benchmark datasets like CIFAR-10/100, limiting the ability to compare its performance with well-established methods.

- Inadequate Comparative Analysis: There seems to be a lack of sufficient comparison with other domain-agnostic methods, such as those proposed by Lee et al. and Wu et al.

- Dataset Limitations: Some datasets used for evaluation, like Kvasir, are not standard benchmarks in their respective domains, which might limit the generalizability of the results.

- Transfer Learning and Scalability: The absence of medium/large scale datasets and transfer learning experiments in the results section obscures the method's benefits for broader SSL tasks.

Given these insights, the paper presents a novel and potentially impactful approach in SSRL, with its simplicity and wide applicability being major strengths. However, the concerns regarding dataset selection, computational efficiency, and the need for more comprehensive comparative and theoretical analyses suggest areas for improvement. Addressing these issues could enhance the paper's contribution to the field and its potential for real-world application.

**Justification For Why Not Higher Score:**

See weaknesses.

**Justification For Why Not Lower Score:**

See strengths.

---

### Decision · Program_Chairs · 2024-01-16

Accept (poster)